# Adaptive Resilient Neural Control of Uncertain Time-Delay Nonlinear CPSs with Full-State Constraints under Deception Attacks

**DOI:** 10.3390/e25060900

**Published:** 2023-06-05

**Authors:** Zhihao Chen, Xin Wang, Ning Pang, Yushan Shi

**Affiliations:** 1WESTA College, Southwest University, Chongqing 400700, China; czh011129@126.com (Z.C.); ningpangswu76@163.com (N.P.); whenchan@163.com (Y.S.); 2College of Electronic and Information Engineering, Southwest University, Chongqing 400700, China

**Keywords:** adaptive resilient neural control, full-state constraints, deception attacks, unknown time-delay, dynamic surface method

## Abstract

This paper focuses on the adaptive control problem of a class of uncertain time-delay nonlinear cyber-physical systems (CPSs) with both unknown time-varying deception attacks and full-state constraints. Since the sensors are disturbed by external deception attacks making the system state variables unknown, this paper first establishes a new backstepping control strategy based on compromised variables and uses dynamic surface techniques to solve the disadvantages of the huge computational effort of the backstepping technique, and then establishes attack compensators to mitigate the impact of unknown attack signals on the control performance. Second, the barrier Lyapunov function (BLF) is introduced to restrict the state variables. In addition, the unknown nonlinear terms of the system are approximated using radial basis function (RBF) neural networks, and the Lyapunov–Krasovskii function (LKF) is introduced to eliminate the influence of the unknown time-delay terms. Finally, an adaptive resilient controller is designed to ensure that the system state variables converge and satisfy the predefined state constraints, all signals of the closed-loop system are semi-globally uniformly ultimately bounded under the premise that the error variables converge to an adjustable neighborhood of origin. The numerical simulation experiments verify the validity of the theoretical results.

## 1. Introduction

The fundamental factor behind the advancement of theoretical approaches is the need in engineering practice. Since nonlinear systems make up the majority of systems in real engineering and cannot be approximated by utilizing linear approaches, stability analysis, and control design for unknown nonlinear systems have recently received a lot of interest from academics across the globe [1,2,3,4]. For nonlinear systems with strict feedback, the backstepping control technique was proposed in [5,6,7,8], which laid the groundwork for resolving nonlinear systems’ control issues. The dynamic surface approach was introduced in [9] to improve the backstepping control strategy and address the problem of “complexity explosion”. This considerably increased the backstepping technique’s range of applications. The design of the system controller is greatly complicated by the presence of unknown nonlinear functions in the nonlinear system. Many studies and tests have demonstrated that the dynamic surface backstepping control method, which is based on the approximation technique, is effective in solving the adaptive control problem of unknown nonlinear systems [10,11,12,13,14,15,16,17,18,19]. The more advanced approaches now available for the system’s unknown nonlinear variables use neural network approximation techniques [10,13,14,16,17] and fuzzy approximation techniques [15,18,19]. With the development of theoretical research, adaptive tracking control in multiple input multiple outputs (MIMO) systems has been accomplished using adaptive neural network approximation approaches [20,21]. Moreover, the time-delay effect is another problem that cannot be disregarded in the adaptive control of nonlinear systems. This issue is prevalent in real-world engineering applications such as magnetic levitation systems, crane systems, network control systems, etc. Therefore, the LKF is utilized in [16,22,23] for nonlinear systems with strict feedback and unknown time-delay to eliminate the effect of the time-delay component and combine with the dynamic surface technique to recursively build the controller to make the system stable. The traditional recursive approach does not work in this case because, as was mentioned in [24], the recursive design strategy used in the aforementioned paper is based on precise state variables, and once the system’s state variables are corrupted by outside attack signals, the system is paralyzed or even experiences a serious failure.

CPSs, which combine computational, networking, and physical components, are multidimensional complex systems. The CPS is made up of a communication network that connects hardware components to create an organism for exchanging information. Real-time data collection and actuator control are the responsibilities of the perception layer, while data transmission and communication are the responsibilities of the network layer. The control layer is in charge of calculating and analyzing the data transmitted by the perception layer to provide instructions to the controller for the subsequent operation. From the above description, we can infer that the network layer of CPS serves as a link between the physical and digital worlds and is also the layer that is most susceptible to hostile external attacks. As a result, improving CPS’s security control is one of the system’s urgent problems.

According to [25], the three main types of network attacks are Deception attacks [24,26,27], Denial-of-service (Dos) attacks [28,29,30], and Replay attacks [31,32]. The deception attack is a way for an outside malicious attacker to change the actual state variable information of the system by injecting error information to achieve their desired damage, such as Stuxnet, etc. The specific way to inject information is to apply additive signals to the CPS sensors and actuators to change the original measurement information and execution commands. Since the deception attack does not change the state information of the system, only the state measured at the control layer is changed by the attack signals, it should be a measurement attack according to the statement in [33].

Some practical examples of deception attacks are considered: in [34], an event-triggered adaptive control strategy for deception attacks on bottleneck sections of high-speed trains is proposed to achieve stable passage of a multi-train system; in [35], the anti-deception attack problem for network-based load frequency control is studied, and the proposed adaptive event-triggered control method effectively trades off communication resources with control performance. In [36], an adaptive control method for linear systems with unknown sensor and actuator attacks is proposed. Ref. [37] discusses adaptive control of a nonlinear CPS against a false data injection attack. The stability control problem of nonlinear strict feedback systems with a lower triangular structure is solved using a recursive control design method based on compromised variables in [24], which solves the issue that state variables corrupted by unknown deception attacks cannot be used in the design of recursive controllers. To explore the adaptive control of a class of switched systems with unknown deception attacks, [38] employs the novel recursive technique used in [24]. Although the aforementioned studies have produced promising experimental results based on certain hypotheses, in actual engineering practice, it is impossible to avoid the various constraint problems connected to them, making the aforementioned techniques ineffective. Therefore, the theoretical studies should take into account the state constraint problem of the system. 

The study of state constraints for nonlinear systems has grown to be a popular topic in this subject since state constraints frequently and significantly impact a system’s performance, stability, and even safety. The issue of a class of nonlinear systems in a strict feedback form subject to output constraints was addressed in earlier research [39]. In [40] and [41], respectively, partial-state and full-state constraint problems for strict feedback systems were explored. In [42], strategies for controlling a class of nonlinear systems with time-varying full-state constraints using adaptive neural network technology were proposed. Ref. [43] proposed an adaptive neural network-based tracking control approach for nonlinear systems with time-delay effects. Based on the aforementioned findings, it can be known that BLF is a useful technique for handling state-constrained issues. It is of great theoretical and practical importance to study nonlinear CPSs with both state constraints and unknown deception attacks because more general nonlinear CPSs are not only constrained by a variety of practical factors but also have their communication network vulnerable to interference from outside malicious attackers. 

Adaptive resilient control of a class of nonlinear systems with full-state constraints under an unknown deception attack is the main topic of this work. This is a new solution to the new problem of adaptive resilient control of nonlinear time-delayed systems with deception attacks and full-state constraints. In security control of CPS, this article effectively accomplishes the anti-attack control of CPS while making the system state satisfy the constraints.

The analysis is predicated on [24], where all sensors are taken to be subject to extra unknown attack signals, and all time-delay terms of the system are taken to be unknown and time-varying. The unknown nonlinear function of the system is estimated using an RBF neural network-based approximation method. The unknown time-delay terms are eliminated from the controller design by introducing LKF. The entire recursive design is based on the compromised variables in the backstepping control method mentioned in [24]. An attack compensator is established in each step to reduce the impact of the unknown deception attacks. The following two points serve as a summary of this paper’s specific contributions:The adaptive control problem of nonlinear systems with full-state constraints and unknown deception attacks is investigated for the first time based on the studies in [24,38], and BLF is introduced for state constraints so that the system can satisfy full-state constraints as well as remain stable under external deception attacks.A novel adaptive resilient recursive control approach based on state variables compromised by deception attack signals is employed for unknown time-varying deception attacks. An attack compensator is created during the recursive process to handle the effects of the unknown deception attacks on the controller design.

This paper’s general organization can be broken down into six sections. The research foundation and the problem this paper is addressing are introduced in this segment, which is the study’s opening section. The system’s mathematical model, together with any underlying presumptions and lemmas, is introduced in Section 2. Section 3 introduces the techniques utilized in the controller’s following recursive design. The controller’s recursive design process and the system’s stability analysis are both covered in Section 4. To confirm that the theory is feasible, Section 5 conducts two simulation tests. The entire paper is concluded in Section 6.

## 2. Problem Formulation

Consider the following mathematical model of an unknown nonlinear time-delay system under deception attacks:(1)$$\left\{\begin{array}{c} {\dot{x}}_{i}(t)=x_{i+1}(t)+f_{i}({\bar{x}}_{i}(t))+h_{i}({\bar{x}}_{i}(t-\tau_{i}(t)))\\ {\dot{x}}_{n}(t)=u+f_{n}({\bar{x}}_{n}(t))+h_{n}({\bar{x}}_{n}(t-\tau_{n}(t)))+\varphi_{a}(t,{\bar{x}}_{n}(t))\\ {\overset{⌣}{x}}_{j}(t)=x_{j}(t)+\varphi_{j,s}(t,x_{j}(t))\\ y(t)=x_{1}(t) \end{array}\right.$$
where i=1,2,…,n−1 represents the number of subsystems the system has and j=1,2,…,n represents the total number of sensors it has, ${\bar{x}}_{j}(t)=[x_{j}(t),x_{j}(t),\ldots ,x_{j}(t)]\in \Omega^{j}$ is the actual state variable while $y(t)=x_{1}(t)\in \Omega$ is taken to be the system’s output, ${\bar{x}}_{j}(t-\tau_{j}(t))=[x_{1}(t-\tau_{1}(t)),x_{2}(t-\tau_{2}(t)),\ldots ,x_{j}(t-\tau_{j}(t))]\in \Omega^{j}$ is the state variable with an unknown time-delay $\tau_{j}(t)$, u∈Ω is the actual control input. $f_{j}({\bar{x}}_{j}(t)):\Omega^{j}\mapsto \Omega$ is an unknown nonlinear continuous function, $h_{j}({\bar{x}}_{j}(t-\tau_{j}(t))):\Omega^{j}\mapsto \Omega$ is an unknown nonlinear continuous function that has an unknown time-delay, $\varphi_{a}(t,{\bar{x}}_{n}(t)):\Omega^{n}\mapsto \Omega$ is an actuator deception attack, $\varphi_{j,s}(t,x_{j}(t))$ is a sensor deception attack, etc. These two types of attacks’ functions are all unknown time-varying continuous. As well, the compromised variable of the system variable $x_{j}(t)$ following the occurrence of the unknown deception attack is defined as ${\overset{⌣}{x}}_{j}(t)$, and Assumption 1 elaborates on its specific mathematical expression.

**Remark** **1.***To ensure the reliability of the system model, refer to [24], the initial state of the system state variable* ${\bar{x}}_{n}(t_{0})=\bar{ℑ}(t_{0}),\;-m\leq t_{0}\leq 0$ *is specified in this paper. For the unknown time-delay* $\tau_{j}(t)$ *mentioned in the system Equation (1), there is* $0<\tau_{j}\leq m_{j}<\infty$ *and* $0<{\dot{\tau }}_{j}\leq m_{j,d}<\infty$*, so* m *can be defined as* $m=\max\{m_{1},m_{2},m_{j},\ldots ,m_{n}\}$*. To guarantee the stability of the system and the model of universal adaptation, this definition describes the beginning moment of the system state and the boundedness of the unknown time-delay term.*

**Remark** **2.***The nonlinear functions* $f_{j}({\bar{x}}_{j}(t))$ *and* $h_{j}({\bar{x}}_{j}(t-\tau_{j}(t)))$ *reflect the nonlinearity and uncertainties of the system shown in (1). Note that since the unknown nonlinear terms of the system need to be approximated by the RBF neural network, it is required that the functions* $f_{j}({\bar{x}}_{j}(t))$ *and* $h_{j}({\bar{x}}_{j}(t-\tau_{j}(t)))$ *are bounded. Of course, if there are unbounded terms in the dynamic equations of the system (1), then the system must be unstable, and it is meaningless to discuss the control method any further.*

**Remark** **3.**
*System (1) represents a class of nonlinear systems with unknown time-delay terms and unknown nonlinear terms. Each of its sensors suffers from an unknown deception attack, and the actuators of the system suffer from the same type of attack. The actual systems mentioned in [23], such as robotic systems and recycling chemical reactors, and networked control systems, as well as electrical networks in [44], may suffer from external malicious attacks in their communication networks, so the problem can be described by the differential equations of the system (1).*


**Control** **Objective*:***
*The central issue of this study is the adaptive resilient control problem for a class of time-delay nonlinear systems with full-state constraints under unknown deception attacks. The established controller makes the system stable, and the state variables satisfy the predetermined constraint limits. In addition, all of the system’s closed-loop signals are uniformly ultimately bounded, and the stability error converges to an adjustable neighborhood of the origin.*


**Assumption 1** **[24,45].***It is clear from the system’s features that the sensor and actuator attack signals are reliant on the system’s state variables. Define the sensor attack signal* $\varphi_{j,s}(t,x_{j}(t))=\delta_{j}(t)x_{j}(t),\;j=1,2,\ldots ,n$ *and actuator attack signal* $\varphi_{a,s}(t,{\bar{x}}_{n}(t))=\phi_{a}(t)q({\bar{x}}_{n}(t))$*, where* $\delta_{j}(t)$ *and* $\phi_{a}(t)$ *are unknown time-varying signals and* $q({\bar{x}}_{n}(t)):\Omega^{n}\mapsto \Omega$ *is an unknown continuous nonlinear function. On this basis, both the unknown time-varying attack signals* $\delta_{j}(t)$ *and* $\phi_{a}(t)$ *are bounded, i.e.,* $\left|\delta_{j}(t)\right|<{\bar{\delta }}_{j}$*,* $\left|{\dot{\delta }}_{j}(t)\right|<{\bar{\delta }}_{j,d}$*, and *$\left|\phi_{a}(t)\right|<{\bar{\phi }}_{a}$*, where* ${\bar{\delta }}_{j}$*,* ${\bar{\delta }}_{j,d}$ *and* ${\bar{\phi }}_{a}$ *are unknown constants.*

**Assumption** **2** **[45].***From Assumption 1, we can know that the compromised variable* ${\overset{⌣}{x}}_{j}(t)=x_{j}(t)+\varphi_{j,s}(t,x_{j}(t)),\;j=1,2,\ldots ,n$ *in the system (1) can be transformed into the following form* ${\overset{⌣}{x}}_{j}(t)=(1+\delta_{j}(t))x_{j}(t).$ *If* $\lambda_{j}(t)=\frac{1}{1+\delta_{j}(t)}$ *is defined, then* $x_{j}(t)$ *can be written as*(2)$$x_{j}(t)=\lambda_{j}(t){\overset{⌣}{x}}_{j}(t).$$Assumption 1 states that $\lambda_{j}(t)$ is a bounded function, and this paper specifies that $\lambda_{j}(t)$ has a positive sign. We have $0<{\underline{\lambda }}_{j}\leq \lambda_{j}(t)\leq {\bar{\lambda }}_{j}<1$ to assume ${\underline{\lambda }}_{j}$ and ${\bar{\lambda }}_{j}$ are unknown constants.

**Remark**  **4.**
*For the two aforementioned assumptions, the following two explanations are provided.*
The model cited in [24,45] is based on Assumption 1, and it is significant to note that the unknown deception attack is introduced into a nonlinear system to cause damage. So, the attack signal must be controllable by the attacker for the attack to have the intended destructive effect. The main prerequisite for a controllable signal is that the signal is bounded because it is generally known that an infinite signal cannot be controlled. As a result, we reasonably assume that the attack signals $\delta_{j}(t)$ and $\phi_{a}(t)$ are bounded.Similar to what was said in [24], the reason we introduced the compromised variable ${\overset{⌣}{x}}_{j}(t)$ is that the impact of unknown deception attacks prevents us from measuring the actual state variable. We should define a variable first, that is, $\varpi_{j}=1+\delta_{j}(t)$. Because if $\varpi_{j}=0$, it also makes ${\overset{⌣}{x}}_{j}(t)=\varpi_{j}x_{j}(t)=0$, which prevents us from using the compromised variable ${\overset{⌣}{x}}_{j}(t)$ as intended, so *Assumption 2* specifies that $\varpi_{i}>0$.


**Lemma** **1 [46].***For any real-valued continuous function* f(x,y) *whose variables are* $x\in \Omega^{m}$*,* $y\in \Omega^{n}$*, respectively, the following inequality holds:*(3)$$\left|f(x,y)\right|\leq g_{1}(x)+g_{2}(y)$$
where $g_{1}(x)\geq 0$ and $g_{2}(y)\geq 0$ are continuous scalar functions.

**Lemma** **2 [47].***Lemma 1 states that nonlinear functions with unknown time-delay terms are handled using the separation method described in [46], that is*(4)$$h_{j}({\bar{x}}_{j}(t-\tau_{j}(t)))\leq \sum_{m=1}^{j}\mu_{j,m}(x_{m}(t-\tau_{m}(t)))$$
where $\mu_{j,m}(\cdot )$ is an unknown and continuous function.

**Definition** **1 [24].***The system stability discussed in this paper refers to the semi-global uniformly ultimately bounded. The definitions are as follows:* β *exists in an adjustable neighborhood of the origin with* β∈(0,b)*, where* b>0 *is an adjusted constant. There also exists a time constant* N*, related to* β *and* c*, i.e.,* N=N(β,c)*, where* c>0 *is also an adjusted constant. There is:* $\left\|{\bar{x}}_{n}(t_{0})\right\|\leq \beta ,\;\forall t\geq t_{0}+N$*, where* $t_{0}$ *is the initial time from Remark 1 and* $\left\|{\bar{x}}_{n}(t_{0})\right\|=\sup_{-m\leq o\leq 0}\left\|{\bar{x}}_{n}(t_{0}+o)\right\|$*.*

## 3. Methodology

### 3.1. RBF Neural Network Approximate Technique

Since RBF neural networks can approximate any nonlinear function with arbitrary accuracy, and they have global approximation capability with fast learning and convergence, RBF neural networks are used in this paper to approximate the unknown nonlinear function in system (1). As mentioned in [47], if the approximation error is given as $\bar{\varepsilon }>0$, for any nonlinear continuous function Q(Z) on a compact set $\Xi_{Z}\in \Omega^{q}$, there exists an RBF neural network that approximates Q(Z), i.e.,
(5)$$Q(Z)=\omega^{*}{}^{⊤}S(Z)+\varepsilon (Z),\;\varepsilon (Z)<\bar{\varepsilon }$$
where $\omega^{*}$ stands for the ideal weight vector of the neural network, which can be defined as $\omega^{*}:=\arg\underset{\omega \in \Omega^{N}}{\min}\{\underset{Z\in \Xi_{Z}}{\sup}\;\left|Q(Z)-\omega^{⊤}S(Z)\right|\}$. N is the number of neurons and ε(Z) is the approximation error. S(Z) is the RBF neural network’s activation function vector and $S(Z)={\left[s_{1}(Z),s_{2}(Z),\ldots ,s_{N}(Z)\right]}^{⊤}\in \Omega^{N}$ is precisely a Gaussian function which are listed as
(6)$$s_{i}(Z)=\exp\left[-\frac{{\left(Z-\varpi_{i}\right)}^{⊤}(Z-\varpi_{i})}{l_{i}^{2}}\right]$$
where $\varpi_{i}={\left[\varpi_{i1},\varpi_{i2},\ldots ,\varpi_{iq}\right]}^{⊤}\in \Omega^{q},\;i=1,2,\ldots ,N$ is the center of the receptive field; $l_{i}$ is the width of the Gaussian function.

**Assumption 3** [48]**.** *There exists an unknown constant* $\bar{W}>0$ *that ensures that the RBF neural network weights are bounded* $\left\|W\right\|<\bar{W}$*.*

**Lemma** **3 [48].***The RBF neural network’s activation function is a bounded Gaussian function, that is* $\left\|S(Z)\right\|\leq \bar{S}$*.*

### 3.2. Full-State Constraints: Barrier Lyapunov Function

In [49], to ensure that the constraints on the system state variables are not violated, the BLF based on the error surface is proposed:(7)$$V_{B}=\frac{1}{2}\log\left(\frac{k_{b}^{2}}{k_{b}^{2}-e^{2}}\right)$$
where $\log\left(\cdot \right)$ stands for the natural logarithm, e is the error surface, and $k_{b}>0$ is the constraint constant. The function value of this function will converge to infinity when the error gets close to the constraint $k_{b}$, guaranteeing that the state constraints are not broken. In the future backstepping recursive design process, this function will be employed as a component of the Lyapunov function to create the Lyapunov function for each subsystem.

**Lemma** **4 [41].***For any positive real number* $k_{b}$ *and the variable* θ∈Ω *satisfying the condition that* $\left|\theta \right|<k_{b}$*, the following inequality holds:*

(8)$$\log\left(\frac{k_{b}^{2p}}{k_{b}^{2p}-\theta^{2p}}\right)<\frac{\theta^{2p}}{k_{b}^{2p}-\theta^{2p}}$$
where p is a positive constant.

## 4. Establishment of the Controller

In this section, a recursive design strategy based on RBF neural networks for the controller of system (1) will be developed so that all variables of the closed-loop system converge, while ensuring the full-state constraints of the system. Inspired by the novel backstepping recursive control strategy in [24], the error surface based on the compromised variables is firstly established, followed by the recursive design of the system controller, and finally, the stability of the system is tested.

Using the Backstepping recursive control design method as a framework, we decompose (1) into n first-order systems and construct Lyapunov functions for each first-order system to obtain the specific form of the virtual controller and to ensure the stability of each first-order system. With the previous n−1 virtual controller forms, we can derive the specific form of the actual controller of the system in the n-th step. In this process, in order to make the form of the virtual controller simple and to keep the system stable, we need to choose the specific form of the appropriate attack compensator and the RBF neural network weight adaptive law. Eventually, the following conclusions were obtained: (9)$$u=-{\hat{\omega }}_{n}^{⊤}S_{n}({\overset{⌣}{Z}}_{n})+\frac{\alpha_{n}-z_{n}}{\nu_{n}}-(2k_{n}+l_{n}+{\left(2k_{n}+{\bar{k}}_{n}\right)}^{2}z_{n}^{2})\gamma_{n}-\frac{2k_{n}\gamma_{n}+{\left(2k_{n}+{\bar{k}}_{n}\right)}^{2}z_{n}^{2}\gamma_{n}}{1+2(k_{b_{n}}^{2}/\lambda_{n}^{2}-{\overset{⌣}{e}}_{n}^{2})}$$
(10)$${\dot{\hat{\omega }}}_{1}=\sigma_{1}({\overset{⌣}{e}}_{1}+\frac{{\overset{⌣}{e}}_{1}}{k_{b_{1}}^{2}/\lambda_{1}^{2}-{\overset{⌣}{e}}_{1}^{2}})S_{1}({\overset{⌣}{Z}}_{1})-\sigma_{1}a_{1}{\hat{\omega }}_{1}$$
(11)$${\dot{\hat{\omega }}}_{j}=\sigma_{j}(\gamma_{j}+\frac{{\overset{⌣}{e}}_{j}}{k_{b_{j}}^{2}/\lambda_{j}^{2}-{\overset{⌣}{e}}_{j}^{2}})S_{j}({\overset{⌣}{Z}}_{j})-\sigma_{j}a_{j}{\hat{\omega }}_{j}$$
(12)$${\dot{\hat{\delta }}}_{j}=(2k_{j}+l_{j}+{\left(2k_{j}+{\bar{k}}_{j}\right)}^{2}z_{j}^{2})z_{j}\gamma_{j}-(2k_{j}+{\bar{k}}_{j})z_{j}^{2}{\hat{\delta }}_{j}-\pi_{j}{\hat{\delta }}_{j}$$
where j=2,3,…,n and $k_{i}>0$, $l_{i}>0$, ${\bar{k}}_{i}>0$, $\sigma_{i}>0$, $a_{i}>0$, $\pi_{i}>0$ are design constants with i=1,2,3,…,n. $\gamma_{j}={\overset{⌣}{x}}_{j}-{\hat{\delta }}_{j}z_{j}$ is an intermediate variable which can avoid using $e_{j}$ in design of controller. $z_{j}$ is the output signal of filter. ${\hat{\delta }}_{j}$ is attack compensator to mitigate the effect of deception attacks and ${\overset{⌣}{Z}}_{1}={\left[{\overset{⌣}{x}}_{1},{\overset{⌣}{e}}_{1}\right]}^{⊤}$, ${\overset{⌣}{Z}}_{i}={\left[{\bar{\overset{⌣}{x}}}_{i}^{⊤},{\overset{⌣}{e}}_{i-1},{\overset{⌣}{e}}_{i},{\overset{⌣}{z}}_{i}\right]}^{⊤}$ are the input vector of neutral networks. (9) is the actual controller of the system, (10) is the weight adaptive law of the RBF neural network in Step 1, (11) is the adaptive law of RBF neural network weights for Step 2 to Step n, and (12) is the attack compensators from Step 2 to Step n. The origin of all the formulas and parameters will be derived in the next two sections.

### 4.1. Error Surface Coordinate Transformation

This study utilizes the dynamic surface technique to enhance the backstepping technique and eliminate the issue of “complexity explosion” generated by it. As a result, the following coordinate transformation is used:(13)$$\left\{\begin{array}{c} e_{1}=x_{1}\\ e_{i}=x_{i}-z_{i}\\ \varsigma_{i}=z_{i}-\alpha_{i} \end{array}\right.$$
where $e_{1}$ and $e_{i},\;i=2,\ldots ,n$ are the error variables, $\alpha_{i}$ is the virtual controller and also the input signal of the first-order low-pass filter, $z_{i}$ is the output signal of the first-order low-pass filter, and $\varsigma_{i}$ is the error between the input and output signals of the first-order low-pass filter. The specific form of the first-order low-pass filter is
(14)$$\nu_{i}{\dot{z}}_{i}+z_{i}=\alpha_{i},\;z_{i}(0)=\alpha_{i}(0)$$
where $\nu_{i}>0$ is the first-order low-pass filter’s inherent constant. The error surfaces based on the compromised variables are
(15)$$\left\{\begin{array}{c} {\overset{⌣}{e}}_{1}={\overset{⌣}{x}}_{1}\\ {\overset{⌣}{e}}_{i}={\overset{⌣}{x}}_{i}-z_{i} \end{array}\right..$$

In the latter, the controller’s recursive design primarily employs (12) in place of (10).

### 4.2. Recursive Design Process of Controller

The recursive design steps of the adaptive resilient controller for nonlinear time-delay systems (1) with full-state constraints will be presented as follows.

#### 4.2.1. The First Step of Backstepping

Step 1: Consider the first error surface $e_{1}=x_{1}$
(16)$${\dot{e}}_{1}=x_{2}+f_{1}(x_{1})+h_{1}(x_{1}(t-\tau_{1})).$$

The Lyapunov Function $V_{1}$ is chosen as
(17)$$V_{1}=\frac{1}{2\lambda_{1}}e_{1}^{2}+\frac{1}{2\sigma_{1}}{\tilde{\omega }}_{1}^{T}{\tilde{\omega }}_{1}+\frac{\lambda_{1}}{2}\log\frac{k_{b_{1}}^{2}}{k_{b_{1}}^{2}-e_{1}^{2}}+\frac{4}{3}V_{L,1}$$
where the Lyapunov–Krasovskii function $V_{L,1}$ is
(18)$$V_{L,1}=\frac{e^{\iota_{1}m_{1}}}{1-m_{1,d}}\int_{t-\tau_{1}(t)}^{t}e^{-\iota_{1}(t-s)}\psi_{1,1}^{2}(x_{1}(s))ds$$
where $\iota_{1}>0$ is an unknown constant, $\sigma_{1}$ is a design constant and $\psi_{1,1}(\cdot )$ is an unknown continuous nonlinear function which is same as $\mu_{1,1}(\cdot )$ defined in (5). Since this paper has the same delay problem as the one studied in [24], the same form of LKF is chosen to solve the delay problem.

Substituting (16) into the time derivative of $V_{1}$ yields
(19)$$\begin{array}{r} {\dot{V}}_{1}=-\frac{{\dot{\lambda }}_{1}}{2\lambda_{1}^{2}}e_{1}^{2}+\frac{1}{\lambda_{1}}e_{1}(x_{2}+f_{1}(x_{1})+h_{1}(x_{1}(t-\tau_{1})))-\frac{1}{\sigma_{1}}{\tilde{\omega }}_{1}^{⊤}{\dot{\hat{\omega }}}_{1}+\frac{4}{3}V_{L,1}^{\prime }+\frac{{\dot{\lambda }}_{1}}{2}\log\frac{k_{b_{1}}^{2}}{k_{b_{1}}^{2}-e_{1}^{2}}\\ +\lambda_{1}\frac{e_{1}}{k_{b_{1}}^{2}-e_{1}^{2}}(x_{2}+f_{1}(x_{1})+h_{1}(x_{1}(t-\tau_{1}))). \end{array}$$

Calculating the first-order derivative of concerning time t and using the boundness of the time-delay term in Remark 1 to amplify the equation, we can obtain $V_{L,1}^{\prime }\leq -\iota_{1}V_{L,1}+p_{1}-\psi_{1,1}^{2}(x_{1}(t-\tau_{1}))$ and $p_{1}=\frac{e^{\iota_{1}m_{1}}}{1-m_{1,d}}\psi_{1,1}^{2}(x_{1})$.

Then, using Lemma 2, we have
(20)$$\frac{1}{\lambda_{1}}e_{1}h_{1}(x_{1}(t-\tau_{1}))+\frac{\lambda_{1}e_{1}}{k_{b_{1}}^{2}-e_{1}^{2}}h_{1}(x_{1}(t-\tau_{1}))+\frac{4}{3}V_{L,1}^{\prime }\leq -\frac{4}{3}\iota_{1}V_{L,1}+\frac{4}{3}p_{1}+\frac{1}{4\lambda_{1}^{2}}e_{1}^{2}+\frac{3\lambda_{1}^{2}e_{1}^{2}}{4(k_{b_{1}}^{2}-e_{1}^{2})}.$$

In the meanwhile, by introducing auxiliary variables $\eta_{1}=\frac{4\lambda_{1}e_{1}p_{1}(k_{b_{1}}^{2}-e_{1}^{2})}{3c_{1}(k_{b_{1}}^{2}-e_{1}^{2}+\lambda_{1}^{2})}$ and using $x_{2}=e_{2}+z_{2}=e_{2}+\alpha_{2}+\varsigma_{2}$ to simplify the inequality (19), we have
(21)$$\begin{array}{r} {\dot{V}}_{1}\leq (\frac{1}{\lambda_{1}}e_{1}+\frac{\lambda_{1}e_{1}}{k_{b_{1}}^{2}-e_{1}^{2}})[e_{2}+\alpha_{2}+\varsigma_{2}-\frac{{\underline{\lambda }}_{1,d}}{2{\bar{\lambda }}_{1}}e_{1}+f_{1}(x_{1})+\frac{1}{4{\underline{\lambda }}_{1}}e_{1}+\frac{3{\bar{\lambda }}_{1}e_{1}}{4(k_{b_{1}}^{2}-e_{1}^{2})}+\eta_{1}]-\frac{4}{3}\iota_{1}V_{L,1}\\ +\frac{4}{3}p_{1}-\frac{4e_{1}^{2}}{3c_{1}}p_{1}+\frac{{\bar{\lambda }}_{1,d}}{2}\log\frac{k_{b_{1}}^{2}}{k_{b_{1}}^{2}-e_{1}^{2}}+(\frac{{\bar{\lambda }}_{1,d}-1}{2})\frac{e_{1}^{2}}{k_{b_{1}}^{2}-e_{1}^{2}}-\frac{1}{\sigma_{1}}{\tilde{\omega }}_{1}^{T}{\dot{\hat{\omega }}}_{1}-\frac{e_{1}^{2}}{k_{b_{1}}^{2}-e_{1}^{2}}. \end{array}$$

There exists an unknown continuous function $Q_{1}(Z_{1})$ that can be expressed as $Q_{1}(Z_{1})=f_{1}(x_{1})+\frac{e_{1}}{4{\underline{\lambda }}_{1}}+\frac{3{\bar{\lambda }}_{1}e_{1}}{4(k_{b_{1}}^{2}-e_{1}^{2})}-\frac{{\underline{\lambda }}_{1,d}}{2{\bar{\lambda }}_{1}}e_{1}+\eta_{1}$ with the input vector $Z_{1}={\left[x_{1},e_{1}\right]}^{⊤}$.

Then, using RBFNN to approximate the unknown nonlinear function
(22)$$Q_{1}(Z_{1})=\omega_{1}^{⊤}S_{1}(Z_{1})+\varepsilon_{1}(Z_{1}).$$

Substituting Equation (22) into inequality (21), we can obtain
(23)$$\begin{array}{r} {\dot{V}}_{1}\leq (\frac{1}{\lambda_{1}}e_{1}+\frac{\lambda_{1}e_{1}}{k_{b_{1}}^{2}-e_{1}^{2}})[e_{2}+\varsigma_{2}+\omega_{1}^{⊤}S_{1}(Z_{1})+\varepsilon_{1}(Z_{1})]+\frac{4}{3}p_{1}-\frac{4e_{1}^{2}}{3c_{1}}p_{1}+\frac{{\bar{\lambda }}_{1,d}}{2}\log\frac{k_{b_{1}}^{2}}{k_{b_{1}}^{2}-e_{1}^{2}}\\ +(\frac{{\bar{\lambda }}_{1,d}-1}{2})\frac{e_{1}^{2}}{k_{b_{1}}^{2}-e_{1}^{2}}-\frac{1}{\sigma_{1}}{\tilde{\omega }}_{1}^{T}{\dot{\hat{\omega }}}_{1}-\frac{4}{3}\iota_{1}V_{L,1}-\frac{e_{1}^{2}}{k_{b_{1}}^{2}-e_{1}^{2}}. \end{array}$$

Next, we can design the virtual controller to ensure system stability
(24)$$\alpha_{2}=-\frac{l_{1}}{\lambda_{1}}e_{1}-\frac{k_{1}}{\lambda_{1}}e_{1}-{\hat{\omega }}_{1}^{⊤}S_{1}({\overset{⌣}{Z}}_{1})=-l_{1}{\overset{⌣}{e}}_{1}-k_{1}{\overset{⌣}{e}}_{1}-{\hat{\omega }}_{1}^{⊤}S_{1}({\overset{⌣}{Z}}_{1})$$
where $l_{1}$ and $k_{1}$ are design constants. They are specified to have a positive sign, and the positive and negative coefficients are reflected in the front of this term.

And with the help of Lemma 3, the inequality (23) can be reduced to the following form:(25)$$\begin{array}{r} {\dot{V}}_{1}\leq (\frac{1}{\lambda_{1}}e_{1}+\frac{\lambda_{1}e_{1}}{k_{b_{1}}^{2}-e_{1}^{2}})(e_{2}+\varsigma_{2})+(\frac{1}{\lambda_{1}}e_{1}+\frac{\lambda_{1}e_{1}}{k_{b_{1}}^{2}-e_{1}^{2}})[-l_{1}{\overset{⌣}{e}}_{1}-k_{1}{\overset{⌣}{e}}_{1}+{\tilde{\omega }}_{1}^{⊤}S_{1}({\overset{⌣}{Z}}_{1})+\chi_{1}]+\frac{4}{3}p_{1}\\ -\frac{4e_{1}^{2}}{3c_{1}}p_{1}+({\dot{\lambda }}_{1}-\frac{1}{2})\log\frac{k_{b_{1}}^{2}}{k_{b_{1}}^{2}-e_{1}^{2}}-\frac{1}{\sigma_{1}}{\tilde{\omega }}_{1}^{T}{\dot{\hat{\omega }}}_{1}-\frac{4}{3}\iota_{1}V_{L,1}-\frac{e_{1}^{2}}{k_{b_{1}}^{2}-e_{1}^{2}} \end{array}$$
where $\chi_{1}=\omega_{1}^{⊤}(S_{1}(Z_{1})-S_{1}({\overset{⌣}{Z}}_{1}))+\varepsilon_{1}(Z_{1})$, and ${\overset{⌣}{Z}}_{1}={\left[{\overset{⌣}{x}}_{1},{\overset{⌣}{e}}_{1}\right]}^{⊤}$ is also the input vector of RBF neutral network composed of compromised variables.

The weight adaptive law of RBFNN is (11), and bringing it into (25), we can get
(26)$$\begin{array}{r} {\dot{V}}_{1}\leq (\frac{1}{\lambda_{1}}e_{1}+\frac{\lambda_{1}e_{1}}{k_{b_{1}}^{2}-e_{1}^{2}})(e_{2}+\varsigma_{2})+(\frac{1}{\lambda_{1}}e_{1}+\frac{\lambda_{1}e_{1}}{k_{b_{1}}^{2}-e_{1}^{2}})(-l_{1}{\overset{⌣}{e}}_{1}+\chi_{1})-\frac{4}{3}\iota_{1}V_{L,1}+\frac{4}{3}p_{1}-\frac{4e_{1}^{2}}{3c_{1}}p_{1}\\ +a_{1}{\tilde{\omega }}_{1}^{T}{\hat{\omega }}_{1}+({\dot{\lambda }}_{1}-\frac{1}{2})\log\frac{k_{b_{1}}^{2}}{k_{b_{1}}^{2}-e_{1}^{2}}-\frac{(k_{1}+1)e_{1}^{2}}{k_{b_{1}}^{2}-e_{1}^{2}}-k_{1}{\overset{⌣}{e}}_{1}^{2}. \end{array}$$

#### 4.2.2. The i-th Step of Backstepping

Step I (i=2,3,…,n−1): Consider the i-th error surface $e_{i}=x_{i}-z_{i}$, the derivative is
(27)$${\dot{e}}_{i}=x_{i+1}+f_{i}({\bar{x}}_{i})+h_{i}({\bar{x}}_{i}(t-\tau_{i}))-{\dot{z}}_{i}.$$

The Lyapunov Function $V_{i}$ in this step is chosen as
(28)$$V_{i}=\frac{1}{2\lambda_{i}}e_{i}^{2}+\frac{1}{2\sigma_{i}}{\tilde{\omega }}_{i}^{T}{\tilde{\omega }}_{i}+\frac{\lambda_{i}}{2}\log\frac{k_{b_{i}}^{2}}{k_{b_{i}}^{2}-e_{i}^{2}}+\frac{4}{3}V_{L,i}+\frac{1}{2}{\tilde{\delta }}_{i}^{2}$$
where ${\tilde{\delta }}_{i}=\delta_{i}-{\hat{\delta }}_{i}$, $\sigma_{i}$ is a design constant. Lyapunov–Krasovskii functional $V_{L,i}$ is
(29)$$V_{L,i}=\sum_{j=1}^{i}\frac{e^{\iota_{i}m_{j}}}{1-m_{j,d}}\int_{t-\tau_{j}(t)}^{t}e^{-\iota_{i}(t-s)}\psi_{i,j}^{2}(x_{j}(s))ds$$
where $\iota_{j}$ is an unknown constant and $\iota_{j}>0$ and $\psi_{i,j}(\cdot )$ is an unknown continuous nonlinear function which is same as $\mu_{i,j}(\cdot )$ defined in (5).

Substituting (27) into the time derivative of $V_{i}$ yields
(30)$$\begin{array}{r} {\dot{V}}_{i}=-\frac{{\dot{\lambda }}_{i}}{2\lambda_{i}^{2}}e_{i}^{2}+\frac{1}{\lambda_{i}}e_{i}(x_{i+1}+f_{i}({\bar{x}}_{i})+h_{i}({\bar{x}}_{i}(t-\tau_{i}))-{\dot{z}}_{i})+\frac{{\dot{\lambda }}_{i}}{2}\log\frac{k_{b_{i}}^{2}}{k_{b_{i}}^{2}-e_{i}^{2}}-\frac{1}{\sigma_{i}}{\tilde{\omega }}_{i}^{⊤}{\dot{\hat{\omega }}}_{i}\\ +\frac{\lambda_{i}e_{i}}{k_{b_{i}}^{2}-e_{i}^{2}}(x_{i+1}+f_{i}({\bar{x}}_{i})+h_{i}({\bar{x}}_{i}(t-\tau_{i}))-{\dot{z}}_{i})+\frac{4}{3}V_{L,i}^{\prime }+{\tilde{\delta }}_{i}({\dot{\delta }}_{i}-{\dot{\hat{\delta }}}_{i}). \end{array}$$

The first-order derivative of $V_{L,i}$ with respect to time t can be written as $V_{L,i}^{\prime }\leq -\iota_{i}V_{L,i}+p_{i}-\sum_{j=1}^{i}\psi_{i,j}^{2}(x_{j}(t-\tau_{j}))$ with $p_{i}=\sum_{j=1}^{i}\frac{e^{\iota_{i}m_{j}}}{1-m_{j,d}}\psi_{i,j}^{2}(x_{j})$.

Similarly, by using Lemma 3, one has
(31)$$\frac{1}{\lambda_{i}}e_{i}h_{i}({\bar{x}}_{i}(t-\tau_{i}))+\frac{\lambda_{i}e_{i}}{k_{b_{i}}^{2}-e_{i}^{2}}h_{i}({\bar{x}}_{i}(t-\tau_{i}))+\frac{4}{3}V_{L,i}^{\prime }\leq -\frac{4}{3}\iota_{i}V_{L,i}+\frac{4}{3}p_{i}+\frac{1}{4\lambda_{i}^{2}}e_{i}^{2}+\frac{3\lambda_{i}^{2}e_{i}^{2}}{4(k_{b_{i}}^{2}-e_{i}^{2})}.$$

So, the derivative of the Lyapunov function $V_{i}$ is
(32)$$\begin{array}{r} {\dot{V}}_{i}\leq \frac{1}{\lambda_{i}}e_{i}[x_{i+1}+f_{i}({\bar{x}}_{i})-\frac{{\dot{\lambda }}_{i}}{2\lambda_{i}}e_{i}-{\dot{z}}_{i}]+\frac{\lambda_{i}e_{i}}{k_{b_{i}}^{2}-e_{i}^{2}}[x_{i+1}+f_{i}({\bar{x}}_{i})-{\dot{z}}_{i}]+\frac{{\dot{\lambda }}_{i}}{2}\log\frac{k_{b_{i}}^{2}}{k_{b_{i}}^{2}-e_{i}^{2}}+\frac{1}{4\lambda_{i}^{2}}e_{i}^{2}\\ +\frac{3\lambda_{i}^{2}e_{i}^{2}}{4{\left(k_{b_{i}}^{2}-e_{i}^{2}\right)}^{2}}+\frac{4}{3}p_{i}-\frac{4}{3}\iota_{i}V_{L,i}-\frac{1}{\sigma_{i}}{\tilde{\omega }}_{i}^{⊤}{\dot{\hat{\omega }}}_{i}+{\tilde{\delta }}_{i}({\dot{\delta }}_{i}-{\dot{\hat{\delta }}}_{i}). \end{array}$$

Auxiliary variable $\eta_{i}=\frac{4\lambda_{i}e_{i}p_{i}(k_{b_{i}}^{2}-e_{i}^{2})}{3c_{i}(k_{b_{i}}^{2}-e_{i}^{2}+\lambda_{i}^{2})}$ is introduced in this step, and in the meanwhile, we add and subtract some auxiliary terms to help us create some desired terms and eliminate some unknown interfering terms. We can obtain
(33)$$\begin{array}{r} {\dot{V}}_{i}\leq (\frac{1}{\lambda_{i}}e_{i}+\frac{\lambda_{i}e_{i}}{k_{b_{i}}^{2}-e_{i}^{2}})[x_{i+1}+f_{i}({\bar{x}}_{i})-\frac{{\underline{\lambda }}_{i,d}}{2{\bar{\lambda }}_{i}}e_{i}+\frac{1}{4{\underline{\lambda }}_{i}}e_{i}+\frac{3{\bar{\lambda }}_{i}e_{i}}{4(k_{b_{i}}^{2}-e_{i}^{2})}+M_{i}-N_{i}+\eta_{i}-{\dot{z}}_{i}]\\ +({\dot{\lambda }}_{i}-\frac{1}{2})\log\frac{k_{b_{i}}^{2}}{k_{b_{i}}^{2}-e_{i}^{2}}+\frac{4}{3}p_{i}-\frac{4e_{i}^{2}}{3c_{i}}p_{i}-\frac{1}{\lambda_{i-1}}e_{i-1}e_{i}-\frac{\lambda_{i-1}e_{i-1}}{k_{b_{i-1}}^{2}-e_{i-1}^{2}}e_{i}+\frac{1}{\lambda_{i}}e_{i}(2k_{i}+{\bar{k}}_{i})z_{i}\delta_{i}\\ -\frac{e_{i}^{2}}{k_{b_{i}}^{2}-e_{i}^{2}}-\frac{4}{3}\iota_{i}V_{L,i}-\frac{1}{\sigma_{i}}{\tilde{\omega }}_{i}^{⊤}{\dot{\hat{\omega }}}_{i}+{\tilde{\delta }}_{i}({\dot{\delta }}_{i}-{\dot{\hat{\delta }}}_{i}) \end{array}$$
where $M_{i}=\frac{\lambda_{i}e_{i}(k_{b_{i}}^{2}-e_{i}^{2})(k_{b_{i-1}}^{2}-e_{i-1}^{2}+\lambda_{i-1}^{2})}{\lambda_{i-1}(k_{b_{i-1}}^{2}-e_{i-1}^{2})(k_{b_{i-1}}^{2}-e_{i-1}^{2}+\lambda_{i}^{2})}$, $N_{i}=\frac{(2k_{i}+{\bar{k}}_{i})(k_{b_{i}}^{2}-e_{i}^{2})z_{i}\delta_{i}}{\lambda_{i}^{2}+(k_{b_{i}}^{2}-e_{i}^{2})}$.

Then, the unknown continuous nonlinear function $Q_{i}(Z_{i})$ can be expressed as $Q_{i}(Z_{i})=f_{i}({\bar{x}}_{i})-\frac{{\underline{\lambda }}_{i,d}}{2{\bar{\lambda }}_{i}}e_{i}+\frac{1}{4{\underline{\lambda }}_{i}}e_{i}+\frac{3{\bar{\lambda }}_{i}e_{i}}{4(k_{b_{i}}^{2}-e_{i}^{2})}+M_{i}-N_{i}+\eta_{i}.$ By employing RBFNN to approximate the above unknown function $Q_{i}(Z_{i})$ as
(34)$$Q_{i}(Z_{i})=\omega_{i}^{⊤}S_{i}(Z_{i})+\varepsilon_{i}(Z_{i})$$
where the input vector is $Z_{i}={\left[{\bar{x}}_{i}^{⊤},e_{i-1},e_{i},z_{i}\right]}^{⊤}$.

Substituting the (34) into (33), we have
(35)$$\begin{array}{r} {\dot{V}}_{i}\leq (\frac{1}{\lambda_{i}}e_{i}+\frac{\lambda_{i}e_{i}}{k_{b_{i}}^{2}-e_{i}^{2}})(e_{i+1}+\alpha_{i+1}+\varsigma_{i+1}+\omega_{i}^{⊤}S_{i}(Z_{i})+\varepsilon_{i}(Z_{i})-{\dot{z}}_{i})+\frac{4}{3}p_{i}-\frac{4e_{i}^{2}}{3c_{i}}p_{i}\\ +({\dot{\lambda }}_{i}-\frac{1}{2})\log\frac{k_{b_{i}}^{2}}{k_{b_{i}}^{2}-e_{i}^{2}}-\frac{1}{\lambda_{i-1}}e_{i-1}e_{i}-\frac{\lambda_{i-1}e_{i-1}}{k_{b_{i-1}}^{2}-e_{i-1}^{2}}e_{i}+\frac{1}{\lambda_{i}}e_{i}(2k_{i}+{\bar{k}}_{i})z_{i}\delta_{i}\\ -\frac{e_{i}^{2}}{k_{b_{i}}^{2}-e_{i}^{2}}-\frac{4}{3}\iota_{i}V_{L,i}-\frac{1}{\sigma_{i}}{\tilde{\omega }}_{i}^{⊤}{\dot{\hat{\omega }}}_{i}+{\tilde{\delta }}_{i}({\dot{\delta }}_{i}-{\dot{\hat{\delta }}}_{i}). \end{array}$$

In the next step, since the state variables $x_{i}$ and the actual error surface $e_{i}$ of the system cannot be measured, the intermediate variable $\gamma_{i}$ is introduced to replace the actual error surface. From (13) and (15), we have
(36)$$\begin{array}{cc} e_{i} & ={\overset{⌣}{x}}_{i}-z_{i}-\delta_{i}x_{i}\\ & ={\overset{⌣}{x}}_{i}-z_{i}-\delta_{i}(x_{i}-z_{i})-({\tilde{\delta }}_{i}+{\hat{\delta }}_{i})z_{i}\\ & ={\overset{⌣}{x}}_{i}-z_{i}-{\hat{\delta }}_{i}z_{i}-\delta_{i}e_{i}-{\tilde{\delta }}_{i}z_{i}. \end{array}$$

So, using the intermediate variable $\gamma_{i}={\overset{⌣}{x}}_{i}-{\hat{\delta }}_{i}z_{i}$, the above equation can be transformed as $e_{i}=\lambda_{i}(\gamma_{i}-{\tilde{\delta }}_{i}z_{i}).$ It is the representation of the system error surface using the attack compensator. The attack compensator is ${\hat{\delta }}_{i}$, which is the estimated value of the unknown deception attacks. In the course of the subsequent analysis, $e_{i}$ with an attack compensator is used to mitigate the impact of the unknown deception attacks on the system stability control. Then, by substituting the new equation into (35), we can obtain
(37)$$\begin{array}{r} {\dot{V}}_{i}\leq (\frac{1}{\lambda_{i}}e_{i}+\frac{\lambda_{i}e_{i}}{k_{b_{i}}^{2}-e_{i}^{2}})(e_{i+1}+\varsigma_{i+1})+(\gamma_{i}-{\tilde{\delta }}_{i}z_{i})[\alpha_{i+1}+\omega_{i}^{⊤}S_{i}(Z_{i})+\varepsilon_{i}(Z_{i})-{\dot{z}}_{i}]-\frac{4}{3}\iota_{i}V_{L,i}\\ +\frac{\lambda_{i}e_{i}}{k_{b_{i}}^{2}-e_{i}^{2}}[\alpha_{i+1}+\omega_{i}^{⊤}S_{i}(Z_{i})+\varepsilon_{i}(Z_{i})-{\dot{z}}_{i}]+({\dot{\lambda }}_{i}-\frac{1}{2})\log\frac{k_{b_{i}}^{2}}{k_{b_{i}}^{2}-e_{i}^{2}}+\frac{4}{3}p_{i}-\frac{4e_{i}^{2}}{3c_{i}}p_{i}\\ -\frac{e_{i}^{2}}{k_{b_{i}}^{2}-e_{i}^{2}}-\frac{1}{\lambda_{i-1}}e_{i-1}e_{i}-\frac{\lambda_{i-1}e_{i-1}}{k_{b_{i-1}}^{2}-e_{i-1}^{2}}e_{i}+\frac{1}{\lambda_{i}}e_{i}(2k_{i}+{\bar{k}}_{i})z_{i}\delta_{i}-\frac{1}{\sigma_{i}}{\tilde{\omega }}_{i}^{⊤}{\dot{\hat{\omega }}}_{i}+{\tilde{\delta }}_{i}({\dot{\delta }}_{i}-{\dot{\hat{\delta }}}_{i}). \end{array}$$

The virtual control law $\alpha_{i+1}$ in this step can be designed as
(38)$$\alpha_{i+1}=-{\hat{\omega }}_{i}^{⊤}S_{i}({\overset{⌣}{Z}}_{i})+\frac{\alpha_{i}-z_{i}}{\nu_{i}}-(2k_{i}+l_{i}+{\left(2k_{i}+{\bar{k}}_{i}\right)}^{2}z_{i}^{2})\gamma_{i}-\frac{\left(2k_{i}\gamma_{i}+{\left(2k_{i}+{\bar{k}}_{i}\right)}^{2}z_{i}^{2}\gamma_{i}\right)}{1+2(k_{b_{i}}^{2}/\lambda_{i}^{2}-{\overset{⌣}{e}}_{i}^{2})}.$$
where $k_{i}>0$, $l_{i}>0$, ${\bar{k}}_{i}>0$ are design constants of the virtual controller.

${\dot{V}}_{i}$ after adding the virtual controller (38) is
(39)$$\begin{array}{r} {\dot{V}}_{i}\leq (\frac{1}{\lambda_{i}}e_{i}+\frac{\lambda_{i}e_{i}}{k_{b_{i}}^{2}-e_{i}^{2}})(e_{i+1}+\varsigma_{i+1})+\frac{\lambda_{i}e_{i}}{k_{b_{i}}^{2}-e_{i}^{2}}[-{\hat{\omega }}_{i}^{⊤}S_{i}({\overset{⌣}{Z}}_{i})-l_{i}\gamma_{i}+\omega_{i}^{⊤}S_{i}(Z_{i})+\varepsilon_{i}(Z_{i})]\\ +(\gamma_{i}-{\tilde{\delta }}_{i}z_{i})[-{\hat{\omega }}_{i}^{⊤}S_{i}({\overset{⌣}{Z}}_{i})-(2k_{i}+l_{i}+{\left(2k_{i}+{\bar{k}}_{i}\right)}^{2}z_{i}^{2})\gamma_{i}+\omega_{i}^{⊤}S_{i}(Z_{i})+\varepsilon_{i}(Z_{i})]\\ +({\dot{\lambda }}_{i}-\frac{1}{2})\log\frac{k_{b_{i}}^{2}}{k_{b_{i}}^{2}-e_{i}^{2}}+\frac{4}{3}p_{i}-\frac{4e_{i}^{2}}{3c_{i}}p_{i}-\frac{1}{\lambda_{i-1}}e_{i-1}e_{i}-\frac{\lambda_{i-1}e_{i-1}}{k_{b_{i-1}}^{2}-e_{i-1}^{2}}e_{i}-\frac{4}{3}\iota_{i}V_{L,i}\\ -\frac{e_{i}^{2}}{k_{b_{i}}^{2}-e_{i}^{2}}+\frac{1}{\lambda_{i}}e_{i}(2k_{i}+{\bar{k}}_{i})z_{i}\delta_{i}-\frac{1}{\sigma_{i}}{\tilde{\omega }}_{i}^{⊤}{\dot{\hat{\omega }}}_{i}+{\tilde{\delta }}_{i}({\dot{\delta }}_{i}-{\dot{\hat{\delta }}}_{i}). \end{array}$$

Considering the auxiliary term $-\frac{k_{i}}{\lambda_{i}^{2}}e_{i}^{2}$, we have the following inequality $\frac{k_{i}}{\lambda_{i}^{2}}e_{i}^{2}\leq 2(k_{i}\gamma_{i}^{2}+k_{i}{\tilde{\delta }}_{i}^{2}z_{i}^{2})$. Using the inequality to simplify (39), one has
(40)$$\begin{array}{r} {\dot{V}}_{i}\leq (\frac{1}{\lambda_{i}}e_{i}+\frac{\lambda_{i}e_{i}}{k_{b_{i}}^{2}-e_{i}^{2}})(e_{i+1}+\varsigma_{i+1})+\gamma_{i}[{\tilde{\omega }}_{i}^{⊤}S_{i}({\overset{⌣}{Z}}_{i})-(l_{i}+{\left(2k_{i}+{\bar{k}}_{i}\right)}^{2}z_{i}^{2})\gamma_{i}+\chi_{i}]-\frac{4}{3}\iota_{i}V_{L,i}\\ +\frac{\lambda_{i}e_{i}}{k_{b_{i}}^{2}-e_{i}^{2}}[{\tilde{\omega }}_{i}^{⊤}S_{i}({\overset{⌣}{Z}}_{i})-l_{i}\gamma_{i}+\chi_{i}]-{\tilde{\delta }}_{i}z_{i}[{\tilde{\omega }}_{i}^{⊤}S_{i}({\overset{⌣}{Z}}_{i})-(2k_{i}+l_{i}+{\left(2k_{i}+{\bar{k}}_{i}\right)}^{2}z_{i}^{2})\gamma_{i}+\chi_{i}]\\ +({\dot{\lambda }}_{i}-\frac{1}{2})\log\frac{k_{b_{i}}^{2}}{k_{b_{i}}^{2}-e_{i}^{2}}+\frac{4}{3}p_{i}-\frac{4e_{i}^{2}}{3c_{i}}p_{i}-\frac{1}{\lambda_{i-1}}e_{i-1}e_{i}-\frac{\lambda_{i-1}e_{i-1}}{k_{b_{i-1}}^{2}-e_{i-1}^{2}}e_{i}+\frac{1}{\lambda_{i}}e_{i}(2k_{i}+{\bar{k}}_{i})z_{i}\delta_{i}\\ -\frac{e_{i}^{2}}{k_{b_{i}}^{2}-e_{i}^{2}}-\frac{1}{\sigma_{i}}{\tilde{\omega }}_{i}^{⊤}{\dot{\hat{\omega }}}_{i}+{\tilde{\delta }}_{i}({\dot{\delta }}_{i}-{\dot{\hat{\delta }}}_{i})-\frac{k_{i}}{\lambda_{i}^{2}}e_{i}^{2}+2k_{i}{\left({\tilde{\delta }}_{i}z_{i}\right)}^{2} \end{array}$$
where $\chi_{i}=\omega_{i}^{⊤}[S_{i}(Z_{i})-S_{i}({\overset{⌣}{Z}}_{i})]+\varepsilon_{i}(Z_{i})$ and ${\overset{⌣}{Z}}_{i}={\left[{\bar{\overset{⌣}{x}}}_{i},{\overset{⌣}{e}}_{i-1},{\overset{⌣}{e}}_{i},z_{i}\right]}^{⊤}$ is also the input vector of RBF neutral network composed of compromised variables.

Weight adaptive law ${\dot{\hat{\omega }}}_{i}$ and attack compensator ${\dot{\hat{\delta }}}_{i}$ at this step can be designed as (11) and (12), respectively. Substituting (11) and (12) into (40) again, we get
(41)$$\begin{array}{r} {\dot{V}}_{i}\leq (\frac{1}{\lambda_{i}}e_{i}+\frac{\lambda_{i}e_{i}}{k_{b_{i}}^{2}-e_{i}^{2}})(e_{i+1}+\varsigma_{i+1})+\gamma_{i}[-l_{i}\gamma_{i}-{\left(2k_{i}+{\bar{k}}_{i}\right)}^{2}z_{i}^{2}\gamma_{i}+\chi_{i}]-{\tilde{\delta }}_{i}z_{i}[{\tilde{\omega }}_{i}^{⊤}S_{i}({\overset{⌣}{Z}}_{i})-2k_{i}\gamma_{i}\\ -l_{i}\gamma_{i}-{\left(2k_{i}+{\bar{k}}_{i}\right)}^{2}z_{i}^{2}\gamma_{i}+\chi_{i}]+\frac{\lambda_{i}e_{i}}{k_{b_{i}}^{2}-e_{i}^{2}}(-l_{i}\gamma_{i}+\chi_{i})+({\dot{\lambda }}_{i}-\frac{1}{2})\log\frac{k_{b_{i}}^{2}}{k_{b_{i}}^{2}-e_{i}^{2}}+\frac{4}{3}p_{i}-\frac{4e_{i}^{2}}{3c_{i}}p_{i}\\ -\frac{1}{\lambda_{i-1}}e_{i-1}e_{i}-\frac{\lambda_{i-1}e_{i-1}}{k_{b_{i-1}}^{2}-e_{i-1}^{2}}e_{i}+\frac{1}{\lambda_{i}}e_{i}(2k_{i}+{\bar{k}}_{i})z_{i}\delta_{i}-\frac{4}{3}\iota_{i}V_{L,i}-{\tilde{\omega }}_{i}^{⊤}(a_{i}{\hat{\omega }}_{i})-\frac{k_{i}}{\lambda_{i}^{2}}e_{i}^{2}+2k_{i}{\left({\tilde{\delta }}_{i}z_{i}\right)}^{2}\\ +{\tilde{\delta }}_{i}[{\dot{\delta }}_{i}-(2k_{i}+l_{i}+{\left(2k_{i}+{\bar{k}}_{i}\right)}^{2}z_{i}^{2})z_{i}\gamma_{i}-(2k_{i}+{\bar{k}}_{i})z_{i}^{2}{\hat{\delta }}_{i}-\pi_{i}{\hat{\delta }}_{i}]+{\bar{k}}_{i}{\tilde{\delta }}_{i}^{2}z_{i}^{2}-{\bar{k}}_{i}{\tilde{\delta }}_{i}^{2}z_{i}^{2}-\frac{e_{i}^{2}}{k_{b_{i}}^{2}-e_{i}^{2}}. \end{array}$$

From $e_{i}=\lambda_{i}(\gamma_{i}-{\tilde{\delta }}_{i}z_{i})$, we could get ${\tilde{\delta }}_{i}z_{i}=\gamma_{i}-\frac{1}{\lambda_{i}}e_{i}$, then, consider the following term and make it into the form we need, $(2k_{i}+{\bar{k}}_{i}){\tilde{\delta }}_{i}^{2}z_{i}^{2}\leq {\left(2k_{i}+{\bar{k}}_{i}\right)}^{2}z_{i}^{2}\gamma_{i}^{2}+\frac{1}{4}\delta_{i}^{2}-(2k_{i}+{\bar{k}}_{i})z_{i}\delta_{i}\frac{e_{i}}{\lambda_{i}}-(2k_{i}+{\bar{k}}_{i}){\tilde{\delta }}_{i}{\hat{\delta }}_{i}z_{i}^{2}$.

Substituting the above inequality into (41), one has
(42)$$\begin{array}{r} {\dot{V}}_{i}\leq (\frac{1}{\lambda_{i}}e_{i}+\frac{\lambda_{i}e_{i}}{k_{b_{i}}^{2}-e_{i}^{2}})(e_{i+1}+\varsigma_{i+1})+\gamma_{i}(-l_{i}\gamma_{i}+\chi_{i})+\frac{\lambda_{i}e_{i}}{k_{b_{i}}^{2}-e_{i}^{2}}(-l_{i}\gamma_{i}+\chi_{i})-\frac{4e_{i}^{2}}{3c_{i}}p_{i}\\ -{\tilde{\delta }}_{i}z_{i}[{\tilde{\omega }}_{i}^{⊤}S_{i}({\overset{⌣}{Z}}_{i})+\chi_{i}]+({\dot{\lambda }}_{i}-\frac{1}{2})\log\frac{k_{b_{i}}^{2}}{k_{b_{i}}^{2}-e_{i}^{2}}-\frac{1}{\lambda_{i-1}}e_{i-1}e_{i}-\frac{4}{3}\iota_{i}V_{L,i}-\frac{\lambda_{i-1}e_{i-1}}{k_{b_{i-1}}^{2}-e_{i-1}^{2}}e_{i}\\ +a_{i}{\tilde{\omega }}_{i}^{⊤}{\hat{\omega }}_{i}-\frac{k_{i}}{\lambda_{i}^{2}}e_{i}^{2}-{\bar{k}}_{i}{\tilde{\delta }}_{i}^{2}z_{i}^{2}+\frac{4}{3}p_{i}+{\tilde{\delta }}_{i}({\dot{\delta }}_{i}+\pi_{i}{\hat{\delta }}_{i})+\frac{1}{4}\delta_{i}^{2}-\frac{e_{i}^{2}}{k_{b_{i}}^{2}-e_{i}^{2}}. \end{array}$$

#### 4.2.3. The n-th Step of Backstepping

Step n: Consider the n-th error surface $e_{n}=x_{n}-z_{n}$
(43)$${\dot{e}}_{n}=u+f_{n}({\bar{x}}_{n})+h_{n}({\bar{x}}_{n}(t-\tau_{n}))+\varphi_{a}(t,{\bar{x}}_{n}(t))-{\dot{z}}_{n}.$$

The Lyapunov Function $V_{n}$ is chosen as
(44)$$V_{n}=\frac{1}{2\lambda_{n}}e_{n}^{2}+\frac{1}{2\sigma_{n}}{\tilde{\omega }}_{n}^{T}{\tilde{\omega }}_{n}+\frac{\lambda_{n}}{2}\log\frac{k_{b_{n}}}{k_{b_{n}}^{2}-e_{n}^{2}}+\frac{4}{3}V_{L,n}+\frac{1}{2}{\tilde{\delta }}_{n}^{2}$$
where ${\tilde{\delta }}_{n}=\delta_{n}-{\hat{\delta }}_{n}$, $\sigma_{n}$ is a design constant. LyapunovKrasovskii function $V_{L,n}$ is
(45)$$V_{L,n}=\sum_{j=1}^{n}\frac{e^{\iota_{n}m_{j}}}{1-m_{j,d}}\int_{t-\tau_{j}(t)}^{t}e^{-\iota_{n}(t-s)}\psi_{n,j}^{2}(x_{j}(s))ds$$
where $\iota_{n}$ is an unknown constant and $\iota_{n}>0$ and $\psi_{n,j}(\cdot )$ is an unknown continuous nonlinear function which is same as $\mu_{n,j}(\cdot )$ defined in (5).

Substituting (43) into the time derivative of $V_{n}$ yields
(46)$$\begin{array}{r} {\dot{V}}_{n}=-\frac{{\dot{\lambda }}_{n}}{2\lambda_{n}^{2}}e_{n}^{2}+\frac{1}{\lambda_{n}}e_{n}(u+f_{n}({\bar{x}}_{n})+h_{n}({\bar{x}}_{n}(t-\tau_{n}))+\varphi_{a}(t,{\bar{x}}_{n})-{\dot{z}}_{n})+\frac{{\dot{\lambda }}_{n}}{2}\log\frac{k_{b_{n}}^{2}}{k_{b_{n}}^{2}-e_{n}^{2}}-\frac{1}{\sigma_{n}}{\tilde{\omega }}_{n}^{⊤}{\dot{\hat{\omega }}}_{n}\\ +\frac{\lambda_{n}e_{n}}{k_{b_{n}}^{2}-e_{n}^{2}}[x_{n+1}+f_{n}({\bar{x}}_{n})+h_{n}({\bar{x}}_{n}(t-\tau_{n}))+\varphi_{a}(t,{\bar{x}}_{n}(t))-{\dot{z}}_{n}]+\frac{4}{3}V_{L,n}^{\prime }+{\tilde{\delta }}_{n}({\dot{\delta }}_{n}-{\dot{\hat{\delta }}}_{n}). \end{array}$$

The first-order derivative of $V_{L,n}$ with respect to time t can be written as $V_{L,n}^{\prime }\leq -\iota_{n}V_{L,n}+p_{n}-\sum_{j=1}^{n}\psi_{n,j}^{2}(x_{j}(t-\tau_{j}))$ with $p_{n}=\sum_{j=1}^{n}\frac{e^{\iota_{n}m_{j}}}{1-m_{j,d}}\psi_{n,j}^{2}(x_{j})$.

Next, as in Step I, using Lemma 3, we obtain
(47)$$\frac{1}{\lambda_{n}}e_{n}h_{n}({\bar{x}}_{n}(t-\tau_{n}))+\frac{\lambda_{n}e_{n}}{k_{b_{n}}^{2}-e_{n}^{2}}h_{n}({\bar{x}}_{n}(t-\tau_{n}))+\frac{4}{3}V_{L,n}^{\prime }\leq -\frac{4}{3}\iota_{n}V_{L,n}+\frac{4}{3}p_{n}+\frac{1}{4\lambda_{n}^{2}}e_{n}^{2}+\frac{3\lambda_{n}^{2}e_{n}^{2}}{4(k_{b_{n}}^{2}-e_{n}^{2})}.$$

The auxiliary variables $\eta_{n}=\frac{4\lambda_{n}e_{n}p_{n}(k_{b_{n}}^{2}-e_{n}^{2})}{3c_{n}(k_{b_{n}}^{2}-e_{n}^{2}+\lambda_{n}^{2})}$, $M_{n}=\frac{\lambda_{n}e_{n}(k_{b_{n}}^{2}-e_{n}^{2})(k_{b_{n-1}}^{2}-e_{n-1}^{2}+\lambda_{n-1}^{2})}{\lambda_{n-1}(k_{b_{n-1}}^{2}-e_{n-1}^{2})(k_{b_{n-1}}^{2}-e_{n-1}^{2}+\lambda_{n}^{2})}$ and $N_{n}=\frac{(2k_{n}+{\bar{k}}_{n})(k_{b_{n}}^{2}-e_{n}^{2})z_{n}\delta_{n}}{\lambda_{n}^{2}+(k_{b_{n}}^{2}-e_{n}^{2})}$ are introduced to the derivative of the Lyapunov function $V_{n}$ to create some ideal terms. So, we have
(48)$$\begin{array}{r} {\dot{V}}_{n}\leq (\frac{1}{\lambda_{n}}e_{n}+\frac{\lambda_{n}e_{n}}{k_{b_{n}}^{2}-e_{n}^{2}})[u+f_{n}({\bar{x}}_{n})+\phi_{a}(t)q({\bar{x}}_{n}(t))-\frac{{\underline{\lambda }}_{n,d}}{2{\bar{\lambda }}_{n}}e_{n}+\frac{1}{4{\bar{\lambda }}_{n}}e_{n}+\frac{3{\bar{\lambda }}_{n}e_{n}}{4(k_{b_{n}}^{2}-e_{n}^{2})}+M_{n}\\ -N_{n}+\eta_{n}-{\dot{z}}_{n}]+({\dot{\lambda }}_{n}-\frac{1}{2})\log\frac{k_{b_{n}}^{2}}{k_{b_{n}}^{2}-e_{n}^{2}}+\frac{4}{3}p_{n}-\frac{4e_{n}^{2}}{3c_{n}}p_{n}-\frac{1}{\lambda_{n-1}}e_{n-1}e_{n}-\frac{\lambda_{n-1}e_{n-1}}{k_{b_{n-1}}^{2}-e_{n-1}^{2}}e_{n}\\ +\frac{1}{\lambda_{n}}e_{n}(2k_{n}+{\bar{k}}_{n})z_{n}\delta_{n}-\frac{4}{3}\iota_{n}V_{L,n}-\frac{1}{\sigma_{n}}{\tilde{\omega }}_{n}^{⊤}{\dot{\hat{\omega }}}_{n}+{\tilde{\delta }}_{n}({\dot{\delta }}_{n}-{\dot{\hat{\delta }}}_{n})-\frac{e_{n}^{2}}{k_{b_{n}}^{2}-e_{n}^{2}}. \end{array}$$

There also exists an unknown continuous function $Q_{n}(Z_{n})$ that can be expressed as $Q_{n}(Z_{n})=f_{n}({\bar{x}}_{n})+\phi_{a}(t)q({\bar{x}}_{n}(t))+\frac{e_{n}}{4{\bar{\lambda }}_{n}}+\frac{3{\bar{\lambda }}_{n}e_{n}}{4(k_{b_{n}}^{2}-e_{n}^{2})}-\frac{{\underline{\lambda }}_{n,d}}{2{\bar{\lambda }}_{n}}e_{n}+\eta_{n}+M_{n}-N_{n}.$ Then the RBFNN technique can be used to approximate this unknown function
(49)$$Q_{n}(Z_{n})=\omega_{n}^{⊤}S_{n}(Z_{n})+\varepsilon_{n}(Z_{n})$$
where the input vector is $Z_{n}={\left[{\bar{x}}_{n}^{⊤},e_{n-1},e_{n},z_{n}\right]}^{⊤}$.

Substituting the (49) into the (48), we have
(50)$$\begin{array}{r} {\dot{V}}_{n}\leq (\frac{1}{\lambda_{n}}e_{n}+\frac{\lambda_{n}e_{n}}{k_{b_{n}}^{2}-e_{n}^{2}})(u+\omega_{n}^{⊤}S_{n}(Z_{n})+\varepsilon_{n}(Z_{n})-{\dot{z}}_{n})+({\dot{\lambda }}_{n}-\frac{1}{2})\log\frac{k_{b_{n}}^{2}}{k_{b_{n}}^{2}-e_{n}^{2}}+\frac{4}{3}p_{n}\\ -\frac{4e_{n}^{2}}{3c_{n}}p_{n}-\frac{1}{\lambda_{n-1}}e_{n-1}e_{n}-\frac{\lambda_{n-1}e_{n-1}}{k_{b_{n-1}}^{2}-e_{n-1}^{2}}e_{n}+\frac{1}{\lambda_{n}}e_{n}(2k_{n}+{\bar{k}}_{n})z_{n}\delta_{n}-\frac{4}{3}\iota_{n}V_{L,n}\\ -\frac{1}{\sigma_{n}}{\tilde{\omega }}_{n}^{⊤}{\dot{\hat{\omega }}}_{n}+{\tilde{\delta }}_{n}({\dot{\delta }}_{n}-{\dot{\hat{\delta }}}_{n})-\frac{e_{n}^{2}}{k_{b_{n}}^{2}-e_{n}^{2}}. \end{array}$$

Similar to the above process, denoting $\gamma_{n}={\overset{⌣}{x}}_{n}-{\hat{\delta }}_{n}z_{n}$, and we have $e_{n}=\lambda_{n}(\gamma_{n}-{\tilde{\delta }}_{n}z_{n})$. So, the derivative is
(51)$$\begin{array}{r} {\dot{V}}_{n}\leq (\gamma_{n}-{\tilde{\delta }}_{n}z_{n})[u+\omega_{n}^{⊤}S_{n}(Z_{n})+\varepsilon_{n}(Z_{n})-{\dot{z}}_{n}]-\frac{1}{\sigma_{n}}{\tilde{\omega }}_{n}^{⊤}{\dot{\hat{\omega }}}_{n}+({\dot{\lambda }}_{n}-\frac{1}{2})\log\frac{k_{b_{n}}^{2}}{k_{b_{n}}^{2}-e_{n}^{2}}\\ +\frac{\lambda_{n}e_{n}}{k_{b_{n}}^{2}-e_{n}^{2}}[u+\omega_{n}^{⊤}S_{n}(Z_{n})+\varepsilon_{n}(Z_{n})-{\dot{z}}_{n}]+\frac{4}{3}p_{n}-\frac{4e_{n}^{2}}{3c_{n}}p_{n}-\frac{1}{\lambda_{n-1}}e_{n-1}e_{n}-\frac{4}{3}\iota_{n}V_{L,n}\\ -\frac{\lambda_{n-1}e_{n-1}}{k_{b_{n-1}}^{2}-e_{n-1}^{2}}e_{n}+\frac{1}{\lambda_{n}}e_{n}(2k_{n}+{\bar{k}}_{n})z_{n}\delta_{n}+{\tilde{\delta }}_{n}({\dot{\delta }}_{n}-{\dot{\hat{\delta }}}_{n})-\frac{e_{n}^{2}}{k_{b_{n}}^{2}-e_{n}^{2}}. \end{array}$$

At this step, we design the actual controller of the system (10) based on the form of the virtual controller (24) as well as (38) in the above steps. The weight adaptive law ${\dot{\hat{\omega }}}_{n}$ and attack compensator ${\dot{\hat{\delta }}}_{n}$ are also designed as (11) and (12), respectively.

Substituting (10), (11), and (12) into (51) yields
(52)$$\begin{array}{r} {\dot{V}}_{n}\leq \gamma_{n}(-l_{n}\gamma_{n}-{\left(2k_{n}+{\bar{k}}_{n}\right)}^{2}z_{n}^{2}\gamma_{n}+\chi_{n})-{\tilde{\delta }}_{n}z_{n}({\tilde{\omega }}_{n}^{⊤}S_{n}({\overset{⌣}{Z}}_{n})+\chi_{n})+\frac{4}{3}p_{n}-\frac{4e_{n}^{2}}{3c_{n}}p_{n}+a_{n}{\tilde{\omega }}_{n}^{⊤}{\hat{\omega }}_{n}\\ +\frac{\lambda_{n}e_{n}}{k_{b_{n}}^{2}-e_{n}^{2}}(-l_{n}\gamma_{n}+\chi_{n})+({\dot{\lambda }}_{n}-\frac{1}{2})\log\frac{k_{b_{n}}^{2}}{k_{b_{n}}^{2}-e_{n}^{2}}+\frac{1}{\lambda_{n}}e_{n}(2k_{n}+{\bar{k}}_{n})z_{n}\delta_{n}-\frac{4}{3}\iota_{n}V_{L,n}-\frac{k_{n}}{\lambda_{n}^{2}}e_{n}^{2}\\ -\frac{1}{\lambda_{n-1}}e_{n-1}e_{n}+{\tilde{\delta }}_{n}[{\dot{\delta }}_{n}+(2k_{n}+{\bar{k}}_{n})z_{n}^{2}{\hat{\delta }}_{n}+\pi_{n}{\hat{\delta }}_{n}]-\frac{\lambda_{n-1}e_{n-1}}{k_{b_{n-1}}^{2}-e_{n-1}^{2}}e_{n}-\frac{e_{n}^{2}}{k_{b_{n}}^{2}-e_{n}^{2}}+2k_{n}{\tilde{\delta }}_{n}^{2}z_{n}^{2}\\ +{\bar{k}}_{n}{\tilde{\delta }}_{n}z_{n}^{2}-{\bar{k}}_{n}{\tilde{\delta }}_{n}z_{n}^{2} \end{array}$$
where $\chi_{n}=\omega_{n}^{⊤}[S_{n}(Z_{n})-S_{n}({\overset{⌣}{Z}}_{n})]+\varepsilon_{n}(Z_{n})$ and ${\overset{⌣}{Z}}_{n}={\left[{\bar{\overset{⌣}{x}}}_{n},{\overset{⌣}{e}}_{n-1},{\overset{⌣}{e}}_{n},z_{n}\right]}^{⊤}$ is also the input vector of RBF neutral network composed of compromised variables.

Similar to Step I, in this step, we can obtain $2k_{n}{\tilde{\delta }}_{n}^{2}z_{n}^{2}+{\bar{k}}_{n}{\tilde{\delta }}_{n}z_{n}^{2}\leq {\left(2k_{n}+{\bar{k}}_{n}\right)}^{2}z_{n}^{2}\gamma_{n}^{2}+\frac{1}{4}\delta_{n}^{2}$$-(2k_{n}+{\bar{k}}_{n})z_{n}\delta_{n}\frac{e_{n}}{\lambda_{n}}-(2k_{n}+{\bar{k}}_{n}){\tilde{\delta }}_{n}{\hat{\delta }}_{n}z_{n}^{2}$.

Then, we have the final form
(53)$$\begin{array}{r} {\dot{V}}_{n}\leq \gamma_{n}(-l_{n}\gamma_{n}+\chi_{n})-{\tilde{\delta }}_{n}z_{n}({\tilde{\omega }}_{n}^{⊤}S_{n}({\overset{⌣}{Z}}_{n})+\chi_{n})+\frac{\lambda_{n}e_{n}}{k_{b_{n}}^{2}-e_{n}^{2}}(-l_{n}\gamma_{n}+\chi_{n})+a_{n}{\tilde{\omega }}_{n}^{⊤}{\hat{\omega }}_{n}\\ +({\dot{\lambda }}_{n}-\frac{1}{2})\log\frac{k_{b_{n}}^{2}}{k_{b_{n}}^{2}-e_{n}^{2}}+{\tilde{\delta }}_{n}({\dot{\delta }}_{n}+\pi_{n}\delta_{n})-\frac{k_{n}}{\lambda_{n}^{2}}e_{n}^{2}+\frac{1}{4}\delta_{n}^{2}-\frac{4}{3}\iota_{n}V_{L,n}+\frac{4}{3}p_{n}-\frac{4e_{n}^{2}}{3c_{n}}p_{n}\\ -\frac{1}{\lambda_{n-1}}e_{n-1}e_{n}-\frac{\lambda_{n-1}e_{n-1}}{k_{b_{n-1}}^{2}-e_{n-1}^{2}}e_{n}-\frac{e_{n}^{2}}{k_{b_{n}}^{2}-e_{n}^{2}}. \end{array}$$

### 4.3. Stability Analysis

The previous section uses a recursive design approach to decompose the n-order system shown in (1) into n first-order systems to design their corresponding virtual controllers separately, and finally the specific form of the actual controller (10) is obtained. In this section, it will be analyzed whether the designed controller can make the whole system stable, and the conclusion is:

**Theorem** **1.**
*Adaptive resilient control with both unknown deception attack and full-state constraint problems is solved for the unknown time-delay nonlinear system illustrated in (1), and the designed control method consists of controllers (24), (38), and (10), weight adaptive laws (11), and attack compensators (12) to guarantee that all system variables are semi-globally uniformly ultimately bounded, while state variables satisfy the predefined constraint limits, and the error variables converge to an adjustable neighborhood of the origin.*


The Proof of Theorem 1

**Proof.** We select the total Lyapunov function V to simplify the analysis that follows.
(54)$$\begin{array}{cc} V & =\sum_{j=1}^{n}V_{j}+\sum_{j=1}^{n-1}\frac{\varsigma_{j+1}^{2}}{2}\\ & =\sum_{j=1}^{n}\left[\frac{1}{2\lambda_{j}}e_{j}^{2}+\frac{1}{2\sigma_{j}}{\tilde{\omega }}_{j}^{⊤}{\tilde{\omega }}_{j}+\frac{\lambda_{j}}{2}\log\frac{k_{b_{j}}^{2}}{k_{b_{j}}^{2}-e_{j}^{2}}+\frac{4}{3}V_{L,j}\right]+\sum_{j=2}^{n}\frac{1}{2}{\tilde{\delta }}_{j}+\sum_{j=1}^{n-1}\frac{1}{2}\varsigma_{j+1}^{2}. \end{array}$$By taking the first-order derivative of the total Lyapunov function V, we obtain
(55)$$\begin{array}{r} \dot{V}\leq \sum_{j=1}^{n}\left[-\frac{k_{j}}{\lambda_{j}^{2}}e_{j}^{2}+(\gamma_{j}+\frac{\lambda_{j}e_{j}}{k_{b_{j}}^{2}-e_{j}^{2}})(-l_{j}\gamma_{j}+\chi_{j})-\frac{4}{3}\iota_{j}V_{L,j}+({\dot{\lambda }}_{j}-\frac{1}{2})\log\frac{k_{b_{j}}^{2}}{k_{b_{j}}^{2}-e_{j}^{2}}+a_{j}{\tilde{\omega }}_{j}^{⊤}{\hat{\omega }}_{j}\right]\\ +\sum_{j=2}^{n}\left[-{\tilde{\delta }}_{j}z_{j}({\tilde{\omega }}_{j}^{⊤}S_{j}({\overset{⌣}{Z}}_{j})+\chi_{j})+{\tilde{\delta }}_{j}({\dot{\delta }}_{j}+\pi_{j}{\hat{\delta }}_{j})-{\bar{k}}_{j}{\tilde{\delta }}_{j}^{2}z_{j}^{2}+\frac{1}{4}\delta_{j}^{2}\right]-\sum_{j=1}^{n}\frac{e_{j}^{2}}{k_{b_{j}}^{2}-e_{j}^{2}}\\ +\frac{4}{3}\sum_{j=1}^{n}\left(p_{j}-\frac{e_{j}^{2}}{c_{j}}p_{j}\right)+\sum_{j=1}^{n-1}\varsigma_{j+1}{\dot{\varsigma }}_{j+1}+\sum_{j=1}^{n-1}\left(\frac{1}{\lambda_{j}}e_{j}+\frac{\lambda_{j}e_{j}}{k_{b_{j}}^{2}-e_{j}^{2}}\right)\varsigma_{j+1}. \end{array}$$It is required to add that ${\overset{⌣}{e}}_{1}=\gamma_{1}$ for (55), and in this work, let $k_{1}>0$ to assure the accuracy of the calculation. Next, simplify some of the terms in (55) using Young’s inequality so that it can take the desired form. They are listed as follows: $\gamma_{j}\chi_{j}\leq \frac{1}{2}l_{j}\gamma_{j}^{2}+\frac{1}{2l_{j}}{\bar{\chi }}_{j}^{2}$, $-\frac{l_{j}\gamma_{j}\lambda_{j}e_{j}}{k_{b_{j}}^{2}-e_{j}^{2}}\leq \frac{{\bar{\lambda }}_{j}e_{j}^{2}}{2(k_{b_{j}}^{2}-e_{j}^{2})}+\frac{1}{2}l_{j}\gamma_{j}^{2}$, $\frac{\lambda_{j}e_{j}\chi_{j}}{k_{b_{j}}^{2}-e_{j}^{2}}\leq \frac{{\bar{\lambda }}_{j}e_{j}^{2}}{2(k_{b_{j}}^{2}-e_{j}^{2})}+\frac{1}{2}{\bar{\chi }}_{j}^{2}$, $a_{j}{\tilde{\omega }}_{j}^{⊤}{\hat{\omega }}_{j}\leq \frac{a_{j}}{2}{\tilde{\omega }}_{j}^{⊤}{\tilde{\omega }}_{j}+2a_{j}{\bar{\omega }}_{j}^{⊤}{\bar{\omega }}_{j}$, $-{\tilde{\delta }}_{j}z_{j}{\tilde{\omega }}_{j}^{⊤}S_{j}({\overset{⌣}{Z}}_{j})\leq \frac{{\bar{k}}_{j}}{2}{\tilde{\delta }}_{j}^{2}z_{j}^{2}+\frac{1}{2{\bar{k}}_{j}}{\bar{S}}_{j}^{2}{\tilde{\omega }}_{j}^{⊤}{\tilde{\omega }}_{j}$, $-{\tilde{\delta }}_{j}z_{j}\chi_{j}\leq \frac{{\bar{k}}_{j}}{2}{\tilde{\delta }}_{j}^{2}z_{j}^{2}+\frac{1}{2{\bar{k}}_{j}}{\bar{\chi }}_{j}^{2}$, ${\tilde{\delta }}_{j}{\dot{\delta }}_{j}\leq {\bar{k}}_{j}{\tilde{\delta }}_{j}^{2}+\frac{1}{4{\bar{k}}_{j}}{\bar{\delta }}_{j,d}^{2}$, $\pi_{j}{\tilde{\delta }}_{j}{\hat{\delta }}_{j}\leq -\frac{\pi_{j}}{2}{\tilde{\delta }}_{j}^{2}+\frac{\pi_{j}}{2}{\bar{\delta }}_{j}^{2}$, $-\frac{2{\bar{\lambda }}_{j}e_{j}^{2}}{k_{b_{j}}^{2}-e_{j}^{2}}\leq -2{\bar{\lambda }}_{j}\log\frac{k_{b_{j}}^{2}}{k_{b_{j}}^{2}-e_{j}^{2}}$. On top of this, we also need to define some constants to help us further simplify the inequality (55), $A_{j}=(\frac{1}{2l_{j}}+\frac{1}{2}){\bar{\chi }}_{j}^{2}+\frac{a_{j}}{2}{\bar{\omega }}_{j}^{⊤}{\bar{\omega }}_{j}$, $B_{j}=\frac{1}{2{\bar{k}}_{j}}{\bar{\chi }}_{j}^{2}+\frac{1}{4{\bar{k}}_{j}}{\bar{\delta }}_{j,d}^{2}+(\frac{1}{4}+\frac{\pi_{j}}{2}){\bar{\delta }}_{j}^{2}$, $a_{j}^{*}=\frac{2k_{j}}{{\bar{\lambda }}_{j}}$, $b_{j}^{*}=-a_{j}\sigma_{j}-\frac{{\bar{S}}_{j}^{2}\sigma_{j}}{{\bar{k}}_{j}}$, $c_{j}^{*}=\frac{4}{{\bar{\lambda }}_{j}}-\frac{4{\underline{\lambda }}_{j,d}}{{\bar{\lambda }}_{j}}-2$, $d_{j}^{*}=-2{\bar{k}}_{j}+\pi_{j}$. The following form can be obtained:(56)$$\begin{array}{r} \dot{V}\leq \sum_{j=1}^{n}\left[-a_{j}^{*}\frac{1}{2\lambda_{1}}e_{1}^{2}-b_{j}^{*}\frac{1}{2\sigma_{j}}{\tilde{\omega }}_{j}^{⊤}{\tilde{\omega }}_{j}-\frac{4}{3}\iota_{j}V_{L,j}-c_{j}^{*}\frac{\lambda_{j}}{2}\log\frac{k_{b_{j}}^{2}}{k_{b_{j}}^{2}-e_{j}^{2}}+A_{j}\right]+\sum_{j=2}^{n}\left[-d_{j}^{*}\frac{1}{2}{\tilde{\delta }}_{j}^{2}+B_{j}\right]\\ +\sum_{j=1}^{n-1}\varsigma_{j+1}({\dot{\varsigma }}_{j+1}+\frac{1}{\lambda_{j}}e_{j}+\frac{\lambda_{j}e_{j}}{k_{b_{j}}^{2}-e_{j}^{2}})+\frac{4}{3}\sum_{j=1}^{n}(1-\frac{e_{j}^{2}}{c_{j}})p_{j}. \end{array}$$The error $\varsigma_{j}$ between the input signal and the output signal of the first-order low-pass filter in (14) is discussed in detail as follows. The ${\dot{\varsigma }}_{j+1}$ mentioned in (56) can be expressed a
(57)$${\dot{\varsigma }}_{j+1}={\dot{z}}_{j+1}-{\dot{\alpha }}_{j+1}=-\frac{\varsigma_{j+1}}{\nu_{j+1}}-{\dot{\alpha }}_{j+1}.$$It is clear that the stability of the system is significantly impacted by the boundedness of ${\dot{\alpha }}_{j+1},\;j=1,2,\ldots ,n-1$, and the following discussion specifically addresses ${\dot{\alpha }}_{j+1}$. When j=1, ${\dot{\alpha }}_{2}=-l_{1}{\dot{\overset{⌣}{e}}}_{1}-k_{1}{\dot{\overset{⌣}{e}}}_{1}-{\dot{\hat{\omega }}}_{1}^{⊤}S_{1}({\overset{⌣}{Z}}_{1})-{\hat{\omega }}_{1}^{⊤}{\dot{S}}_{1}({\overset{⌣}{Z}}_{1}){\dot{\overset{⌣}{Z}}}_{1}$; and when j>1, ${\dot{\alpha }}_{j+1}=-{\dot{\hat{\omega }}}_{j}^{⊤}S_{j}({\overset{⌣}{Z}}_{j})-{\hat{\omega }}_{j}^{⊤}{\dot{S}}_{j}({\overset{⌣}{Z}}_{j}){\dot{\overset{⌣}{Z}}}_{j}+\frac{{\dot{\varsigma }}_{j}}{\nu_{j}}-(2k_{j}+l_{j}){\dot{\gamma }}_{j}-2{\left(2k_{i}+{\bar{k}}_{i}\right)}^{2}{\dot{z}}_{i}z_{i}\gamma_{i}-{\left(2k_{i}+{\bar{k}}_{i}\right)}^{2}z_{i}^{2}{\dot{\gamma }}_{i}-{\dot{\vartheta }}_{j}$, where $\vartheta_{j}=\frac{2k_{j}\gamma_{j}+{\left(2k_{j}+{\bar{k}}_{j}\right)}^{2}z_{j}^{2}\gamma_{j}}{1+2k_{b_{j}}^{2}/\lambda_{j}^{2}-2{\overset{⌣}{e}}_{j}^{2}}$. For ease of representation, we presume the existence of a continuous function $\Theta_{j+1}\left(\cdot \right)$, one has
(58)$${\dot{\alpha }}_{j+1}=\Theta_{j+1}({\bar{\omega }}_{j},{\bar{e}}_{j},{\bar{\alpha }}_{j},{\bar{\varsigma }}_{j},{\bar{\delta }}_{j})$$
where ${\bar{\omega }}_{j}={\left[{\hat{\omega }}_{1},{\hat{\omega }}_{2},\ldots ,{\hat{\omega }}_{n},{\dot{\hat{\omega }}}_{1},{\dot{\hat{\omega }}}_{2},\ldots ,{\dot{\hat{\omega }}}_{n}\right]}^{⊤}$, ${\bar{\overset{⌣}{e}}}_{j}={\left[{\overset{⌣}{e}}_{1},{\overset{⌣}{e}}_{2},\ldots ,{\overset{⌣}{e}}_{j},{\dot{\overset{⌣}{e}}}_{1},{\dot{\overset{⌣}{e}}}_{2},\ldots ,{\dot{\overset{⌣}{e}}}_{j}\right]}^{⊤}$, ${\bar{\alpha }}_{j}={\left[\alpha_{2},\ldots ,\alpha_{j},{\dot{\alpha }}_{2},\ldots ,{\dot{\alpha }}_{j}\right]}^{⊤}$, ${\bar{\varsigma }}_{j}={\left[\varsigma_{2},\ldots ,\varsigma_{j},{\dot{\varsigma }}_{1},\ldots ,{\dot{\varsigma }}_{j}\right]}^{⊤}$, ${\bar{\delta }}_{j}={\left[\delta_{2},\ldots ,\delta_{j},{\dot{\delta }}_{2},\ldots ,{\dot{\delta }}_{j}\right]}^{⊤}$. There exists an unknown constant ${\bar{\Theta }}_{j+1}>0$ such that $\Theta_{j+1}({\bar{\omega }}_{j},{\bar{e}}_{j},{\bar{\alpha }}_{j},{\bar{\varsigma }}_{j},{\bar{\delta }}_{j})\leq {\bar{\Theta }}_{j+1}$, so that we have
(59)$$\begin{array}{c} \varsigma_{j+1}({\dot{\varsigma }}_{j+1}+\frac{1}{\lambda_{j}}e_{j}+\frac{\lambda_{j}e_{j}}{k_{b_{j}}^{2}-e_{j}^{2}})=-\frac{\varsigma_{j+1}^{2}}{\nu_{j+1}}+G_{j+1}\varsigma_{j+1}\\ \;\leq (-\frac{1}{\nu_{j+1}}+{\bar{k}}_{j+1})\varsigma_{j+1}^{2}+\frac{1}{4{\bar{k}}_{j+1}}{\bar{G}}_{j+1}^{2} \end{array}$$
where $G_{j+1}=\Theta_{j+1}+\frac{1}{\lambda_{j+1}}e_{j}\varsigma_{j+1}+\frac{\lambda_{j}e_{j}}{k_{b_{j}}^{2}-e_{j}^{2}}$. It can be deduced from the boundedness of $\Theta_{j+1}$ that $G_{j+1}\leq {\bar{G}}_{j+1}$, and ${\bar{G}}_{j+1}$ is an unknown constant. Subsequently, by substituting (59) into (56), we can obtain
(60)$$\begin{array}{r} \dot{V}\leq -\sum_{j=1}^{n}\left[a_{j}^{*}\frac{1}{2\lambda_{1}}e_{1}^{2}+b_{j}^{*}\frac{1}{2\sigma_{j}}{\tilde{\omega }}_{j}^{⊤}{\tilde{\omega }}_{j}+\frac{4}{3}\iota_{j}V_{L,j}+c_{j}^{*}\frac{\lambda_{j}}{2}\log\frac{k_{b_{j}}^{2}}{k_{b_{j}}^{2}-e_{j}^{2}}+A_{j}\right]\\ -\sum_{j=2}^{n}\left(d_{j}^{*}\frac{1}{2}{\tilde{\delta }}_{j}^{2}+e_{j}^{*}\frac{1}{2}\varsigma_{j+1}^{2}\right)+\sum_{j=2}^{n}\left(B_{j}+C_{j}\right)+\frac{4}{3}\sum_{j=1}^{n}\left(1-\frac{e_{j}^{2}}{c_{j}}\right) \end{array}$$
where $e_{j}^{*}=2(\frac{1}{\nu_{j+1}}-{\bar{k}}_{j+1})$, $C_{j}=\frac{1}{4{\bar{k}}_{j+1}}{\left|{\bar{G}}_{j+1}\right|}^{2}$.Simplifying (59) again by letting $\rho =\underset{j=1,2,\ldots ,n}{\min}\{a_{j}^{*},b_{j}^{*},c_{j}^{*},d_{j}^{*},e_{j}^{*},\frac{2}{3}\iota_{j}\}$, $m=\sum_{j=1}^{n}A_{j}+\sum_{j=2}^{n}\left(B_{j}+C_{j}\right)$, one obtains
(61)$$\dot{V}\leq -\rho V+m+2\sum_{j=1}^{n}(1-\frac{e_{j}^{2}}{c_{j}})p_{j}.$$It is obvious that just the sign of term $2\sum_{j=1}^{n}(1-\frac{e_{j}^{2}}{c_{j}})p_{j}$ in (60) cannot be determined, which makes our stability analysis more difficult.
(62)$$2\sum_{j=1}^{n}(1-\frac{e_{j}^{2}}{c_{j}})p_{j}=2\sum_{j=1}^{n}(\frac{c_{j}-e_{j}^{2}}{c_{j}})p_{j}.$$In accordance with (62), we divide into three cases and take into account the magnitude of $2\sum_{j=1}^{n}(1-\frac{e_{j}^{2}}{c_{j}})p_{j}$ in each case: the first case is $\left|e_{j}\right|\leq \sqrt{c_{j}}$, i.e., $2\sum_{j=1}^{n}(1-\frac{e_{j}^{2}}{c_{j}})p_{j}\geq 0$; the second case is $\left|e_{j}\right|>\sqrt{c_{j}}$, i.e., $2\sum_{j=1}^{n}(1-\frac{e_{j}^{2}}{c_{j}})p_{j}<0$; for the third instance, first define the set $\Xi_{\upsilon }:=\{e_{\upsilon }|\;\left|e_{\upsilon }\right|\leq \sqrt{c_{\upsilon }}\}$, $\Xi_{ο}:=\{e_{ο}|\;\left|e_{ο}\right|\leq \sqrt{c_{ο}}\}$, and $\Xi_{\upsilon }\cup \Xi_{ο}=\Xi$, which is the combination of the first two cases, i.e., $e_{\upsilon }\in \Xi_{\upsilon },\;e_{ο}\in \Xi_{ο}$.Case 1. ($e_{\upsilon }\in \Xi_{\upsilon },\;\upsilon =1,2,\ldots ,n$): In this case, $2\sum_{j=1}^{n}(1-\frac{e_{j}^{2}}{c_{j}})p_{j}\geq 0$. We analyze the stability of each variable step by step. First, when j=1, $\left|e_{1}\right|\leq \sqrt{c_{1}}$ is bounded, and according to (2) and Assumption 1, we can learn that ${\overset{⌣}{e}}_{1}$, $x_{1}$, and ${\overset{⌣}{x}}_{1}$ are bounded, which means ${\hat{\omega }}_{1}$ is bounded, making the virtual controller $\alpha_{2}$ bounded. Second, when j=2, $\left|e_{2}\right|\leq \sqrt{c_{2}}$ is bounded, and according to Assumption 1, we can learn that ${\overset{⌣}{e}}_{2}$ is bounded; additionally, it can be deduced from the boundedness of $\alpha_{2}$ that its filter signal $z_{2}$ is also bounded; when combined with the boundedness of $z_{2}$, ${\overset{⌣}{e}}_{2}$ and (13) it can also be deduced that $x_{2}$, ${\hat{\omega }}_{2}$, and ${\hat{\delta }}_{2}$ are also bounded, indicating that $\alpha_{3}$ is bounded. When j=3, according to $\left|e_{3}\right|\leq \sqrt{c_{3}}$ is bounded, it can be known ${\overset{⌣}{e}}_{3}$ is bounded, according to $\alpha_{3}$ is bounded, it can be known $z_{3}$ is bounded, and the same procedure as when j=2, it is possible to determine that $x_{3}$, ${\hat{\omega }}_{3}$, ${\hat{\delta }}_{3}$, and $\alpha_{4}$ are bounded. At j=n, which is reached by repeatedly doing the aforementioned process, it is known that the closed-loop system’s actual controller, u, and all of its signals are bounded.Case 2. ($e_{ο}\in \Xi_{ο},\;ο=1,2,\ldots ,n$): In this case, $2\sum_{j=1}^{n}(1-\frac{e_{j}^{2}}{c_{j}})p_{j}<0$, so (60) can be rewritten in the following form:(63)$$\dot{V}\leq -\rho V+m.$$By solving the differential inequality shown in (62) we can obtain
(64)$$V=[V(0)-\frac{m}{\rho }]e^{-\rho t}+\frac{m}{\rho }.$$When (54) and (63) are combined, $\sum_{j=1}^{n}\frac{1}{2\lambda_{j}}e_{j}^{2}\leq [V(0)-\frac{m}{\rho }]e^{-\rho t}+\frac{m}{\rho }$ is obviously obtained. We get $\left\|e\right\|\leq 2\lambda_{m}[V(0)-\frac{m}{\rho }]e^{-\rho t}+2\lambda_{m}\frac{m}{\rho }$ by setting $\lambda_{m}=\max\{\lambda_{1},\lambda_{2},\ldots ,\lambda_{n}\}$ and $\bar{e}={\left[e_{1},e_{2},\ldots ,e_{n}\right]}^{⊤}$. The right-hand side of the inequality is a decreasing function with time, so that $\left\|e\right\|\leq 2\lambda_{m}\frac{m}{\rho }$ when t→∞. The magnitude of the error variable $e_{j}$ depends on the parameters m and ρ. We can see that all of the closed-loop system’s signals are bounded, and the error may be reduced arbitrarily slightly by selecting the proper parameters.Case 3. ($e_{\upsilon }\in \Xi_{\upsilon },\;e_{ο}\in \Xi_{ο},\;\upsilon \neq ο$): In the calculation of the total Lyapunov function of the system, the summation operation makes some terms cancel each other, however, in this instance, the straightforward cumulative elimination is obviously erroneous, necessitating a thorough treatment of these terms. When $e_{\upsilon }\in \Xi_{\upsilon },\;e_{ο}\in \Xi_{ο},\;\upsilon =ο+1$, the term $-\frac{1}{\lambda_{ο}}e_{ο}e_{\upsilon }-\frac{\lambda_{ο}e_{ο}}{k_{b_{ο}}^{2}-e_{ο}^{2}}e_{\upsilon }$ is treated differently than before, we do the following for the term: $-\frac{1}{\lambda_{ο}}e_{ο}e_{\upsilon }-\frac{\lambda_{ο}e_{ο}}{k_{b_{ο}}^{2}-e_{ο}^{2}}e_{\upsilon }\leq \frac{1}{\lambda_{ο}^{2}}e_{ο}^{2}+\frac{1}{4}e_{\upsilon }^{2}-\frac{e_{ο}^{2}}{k_{b_{ο}}^{2}-e_{ο}^{2}}-\frac{1}{4}\lambda_{ο}^{2}e_{\upsilon }^{2}$, and then using Lemma 4 we can get $-\frac{1}{\lambda_{ο}}e_{ο}e_{\upsilon }-\frac{\lambda_{ο}e_{ο}}{k_{b_{ο}}^{2}-e_{ο}^{2}}e_{\upsilon }\leq \frac{1}{\lambda_{ο}^{2}}e_{ο}^{2}-\log\frac{k_{b_{ο}}^{2}}{k_{b_{ο}}^{2}-e_{ο}^{2}}-(\frac{1}{4}{\underline{\lambda }}_{ο}^{2}-\frac{1}{4})c_{\upsilon }$. Next, we slightly modify the constant value set in the previous section, i.e., $a_{j}^{**}=\frac{2k_{j}}{{\bar{\lambda }}_{j}}-1$, $c_{j}^{**}=\frac{4}{{\bar{\lambda }}_{j}}-\frac{4{\underline{\lambda }}_{j,d}}{{\bar{\lambda }}_{j}}-4$, $\rho^{*}=\underset{j=1,2,\ldots ,n}{\min}\{a_{j}^{**},b_{j}^{*},c_{j}^{**},d_{j}^{*},e_{j}^{*},\frac{2}{3}\iota_{j}\}$, $m^{*}=\sum_{j=1}^{n}A_{j}+\sum_{j=2}^{n}\left(B_{j}+C_{j}\right)-\sum_{ο\in \Gamma_{ο},\upsilon \in \Gamma_{\upsilon }}(\frac{1}{4}{\underline{\lambda }}_{ο}^{2}-\frac{1}{4})c_{\upsilon }$. The above inequality (63) can be expressed as $\dot{V}\leq -\rho^{*}V+m^{*}$. Repeating the analysis steps in Case 2, we may draw the conclusion that the closed-loop system’s signals all converge and that the size of the chosen parameters determines the width of the error variable’s convergence domain.□

By proving cases 1–3, the proof of Theorem 1 is finished.Figure 1 is the block diagram of the method proposed in this paper. For a class of unknown nonlinear systems under deception attacks, we use the Backstepping technique as a framework to design a controller u based on Lyapunov stability theory that allows the system to maintain stability while its state variables satisfy a predetermined full-state constraint. 

**Remark** **5.***The parameters selected in the design process above are only sufficient conditions for system stability. As analyzed in Case 2 above, under the premise of ensuring system stability, appropriately increasing* ρ *or decreasing* m *will theoretically reduce the size of the error variable to a certain extent, but it does not mean that the larger the value of* ρ *is, the better, or the smaller the value of* m *is, the better, and the optimal value needs to be further determined in practice*.

**Remark** **6.***The method designed in this paper only guarantees that the system remains stable under the condition that the constraint is satisfied, but does not consider the time for the system to reach stability. As analyzed in the above three cases, the system reaches the steady state when time* t→∞*, but for a practical system device, it is obvious that the shorter the time t to reach stability, the better. This is the limitation of this paper and an urgent problem for the next research work.*

**Remark** **7.***Anti-attack control of CPS is one of the important branches of security control, and the purpose of this article is to keep the system stable, and the system state variables satisfy the constraint limits even when the system is subject to full-state constraints and external unknown deception attacks. I believe that applying it to the control example of a bottleneck section of a high-speed trains systems studied in* [34] *will yield good results.*

**Remark** **8.**
*The following directions can be pursued in the follow-up research process:*

*Without concentrating on the system’s convergence time, the stability of the system is simply ensured in this study. The problem of finite-time adaptive resilient control of nonlinear time-delay systems with actuator failure and error data injection attacks is addressed in [50]. This finite-time resilient control method can be used in the next stage of research to accelerate the convergence speed of the system for nonlinear systems with both unknown deception attacks and full-state constraints.*

*Standing in the perspective of saving resources to consider, event-triggered control methods are proposed for unknown nonlinear systems in [51], but relatively little research has been done on adaptive event-triggered control for nonlinear systems with both unknown deception attacks and full-state constraints, which is a very promising research direction.*

*In this study, all sensors and actuators are presupposed to be affected by unknown deception attacks, although in real-world engineering, this is extremely uncommon and frequently only some of the actuators and sensors are exposed to unknown deception attacks. Therefore, taking into account the system’s resource usage and execution efficiency, a detection and estimation method for deception attacks on the actuators of nonlinear systems is proposed in [52]. Our next step can be to develop an attack detection method based on this approach.*



## 5. Simulation

In this section, two specific examples will be provided to verify the rationality and validity of the theory. The first example is a numerical example and the second example is a practical case, a chemical reactor recovery system. It should be added that the simulation examples in this article are implemented using MATLAB software, without calling any library functions or toolboxes, and the simulation experiments are completed using only script files (.m).

**Example** **1.***To test the validity of the aforementioned theory, simulation experiments are conducted on a numerical example in this section. A third-order nonlinear system is created as shown below:*(65)$$\left\{\begin{array}{c} {\dot{x}}_{1}=x_{2}+f_{1}(x_{1}(t))+h_{1}(x_{1}(t-\tau_{1}(t)))\\ {\dot{x}}_{2}=x_{3}+f_{2}({\bar{x}}_{2}(t))+h_{2}({\bar{x}}_{2}(t-\tau_{2}(t)))\\ {\dot{x}}_{3}=u+\varphi_{a}(t,{\bar{x}}_{3}(t))+f_{3}({\bar{x}}_{3}(t))+h_{3}({\bar{x}}_{3}(t-\tau_{3}(t)))\\ {\overset{⌣}{x}}_{j}=x_{j}+\varphi_{j,s}(t,x_{j}(t)) \end{array}\right.$$*where the unknown nonlinear function without the time-delay term and the unknown nonlinear function with the time-delay term are*
 $f_{1}(x_{1}(t))=x_{1}^{2}(t)$
*,*
 $h_{1}(x_{1}(t-\tau_{1}(t)))=x_{1}(t-\tau_{1}(t))\sin(x_{1}(t-\tau_{1}(t)))$
*;*
 $f_{2}({\bar{x}}_{2}(t))=\exp(-0.1x_{1}(t))\sin(x_{2}(t))+\sin(x_{1}\left(t\right))\sin(x_{2}(t))$
*,*
 $h_{2}({\bar{x}}_{2}(t-\tau_{2}(t)))=\sin(x_{2}(t-\tau_{2}(t)))$
*;*
 $f_{3}({\bar{x}}_{3}(t))=x_{1}(t)x_{3}(t)\cos(x_{2}(t))+\sin(x_{3}(t))\cos(x_{2}(t))$
*,*
 $h_{3}({\bar{x}}_{3}(t-\tau_{3}(t)))=x_{1}(t-\tau_{1}(t))\exp(-0.5x_{3}(t-\tau_{3}(t)))+\sin(x_{3}(t-\tau_{3}(t)))x_{2}(t-\tau_{2}(t))$
*; unknown sensor deception attack signals are*
 $\varphi_{1,s}(t,x_{1})=(1+\sin t)x_{1}$
*,*
 $\varphi_{2,s}(t,x_{2})=(1+\sin t)x_{2}$
*,*
 $\varphi_{3,s}(t,x_{3})=(1+\cos t)x_{3}$
*; unknown actuator attack signal is*
 $\varphi_{a}(t,{\bar{x}}_{3})=\cos(1.5t)x_{1}x_{3}$
*. The unknown time-delay terms are*
 $\tau_{1}(t)=0.5(3+\cos t)$
*,*
 $\tau_{2}(t)=0.5(3+\cos t)$
*,*
 $\tau_{3}(t)=0.3(4+\sin t)$
*. According to the controller designed above, the resilient control strategy is*
(66)$$\alpha_{2}=-l_{1}{\overset{⌣}{e}}_{1}-k_{1}{\overset{⌣}{e}}_{1}-{\hat{\omega }}_{1}^{⊤}S_{1}({\overset{⌣}{Z}}_{1})$$
(67)$$\alpha_{3}=-{\hat{\omega }}_{2}^{⊤}S_{2}({\overset{⌣}{Z}}_{2})+\frac{\alpha_{2}-z_{2}}{\nu_{2}}-(2k_{2}+l_{2}+{\left(2k_{2}+{\bar{k}}_{2}\right)}^{2}z_{2}^{2})\gamma_{2}-\frac{2k_{2}\gamma_{2}+{\left(2k_{2}+{\bar{k}}_{2}\right)}^{2}z_{2}^{2}\gamma_{2}}{1+2(k_{b_{2}}^{2}/\lambda_{2}^{2}-{\overset{⌣}{e}}_{2}^{2})}$$
(68)$$u=-{\hat{\omega }}_{3}^{⊤}S_{3}({\overset{⌣}{Z}}_{3})+\frac{\alpha_{3}-z_{3}}{\nu_{3}}-(2k_{3}+l_{3}+{\left(2k_{3}+{\bar{k}}_{3}\right)}^{2}z_{3}^{2})\gamma_{3}-\frac{2k_{3}\gamma_{3}+{\left(2k_{3}+{\bar{k}}_{3}\right)}^{2}z_{3}^{2}\gamma_{3}}{1+2(k_{b_{3}}^{2}/\lambda_{3}^{2}-{\overset{⌣}{e}}_{3}^{2})}$$
(69)$${\dot{\hat{\omega }}}_{1}=\sigma_{1}({\overset{⌣}{e}}_{1}+\frac{{\overset{⌣}{e}}_{1}}{k_{b_{1}}^{2}/\lambda_{1}^{2}-{\overset{⌣}{e}}_{1}^{2}})S_{1}({\overset{⌣}{Z}}_{1})-\sigma_{1}a_{1}{\hat{\omega }}_{1}$$
(70)$${\dot{\hat{\omega }}}_{i}=\sigma_{i}(\gamma_{i}+\frac{{\overset{⌣}{e}}_{i}}{k_{b_{i}}^{2}/\lambda_{i}^{2}-{\overset{⌣}{e}}_{i}^{2}})S_{i}({\overset{⌣}{Z}}_{i})-\sigma_{i}a_{i}{\hat{\omega }}_{i},\;\;i=2,3$$
(71)$${\dot{\hat{\delta }}}_{i}=(2k_{i}+l_{i}+{\left(2k_{i}+{\bar{k}}_{i}\right)}^{2}z_{i}^{2})z_{i}\gamma_{i}-(2k_{i}+{\bar{k}}_{i})z_{i}^{2}{\hat{\delta }}_{i}-\pi_{i}{\hat{\delta }}_{i},\;\;i=2,3.$$*The initial values of the system state variables are selected to be* ${\bar{x}}_{3}={\left[-0.5,0.6,0\right]}^{⊤}$ *for the system described in (65), and their coefficients are* $k_{1}=k_{2}=0.5$*,* $k_{3}=0.15$*,* $l_{1}=0.01$*,* $l_{2}=0.3$*,* $l_{3}=2.5$*,* ${\bar{k}}_{2}=0.2$*,* ${\bar{k}}_{3}=0.08$*,* $\nu_{2}=0.003$*,* $\nu_{3}=0.002$*,* $\sigma_{1}=1.2$*,* $\sigma_{2}=1$*,* $\sigma_{3}=0.9$*,* $a_{1}=0.008$*,* $a_{2}=0.01$*,* $a_{3}=0.01$*,* $\pi_{2}=1$*,* $\pi_{3}=0.8$*,* $k_{b_{1}}=k_{b_{2}}=k_{b_{3}}=3$*. Neural networks* ${\hat{\omega }}_{1}^{⊤}S_{1}({\overset{⌣}{Z}}_{1})$*,* ${\hat{\omega }}_{2}^{⊤}S_{2}({\overset{⌣}{Z}}_{2})$*, and* ${\hat{\omega }}_{3}^{⊤}S_{3}({\overset{⌣}{Z}}_{3})$ *have* $N_{1}=3$*,* $N_{2}=12$*, and* $N_{3}=48$ *neurons, respectively, and* $l_{1,i}=1.5,\;i=1,2,\ldots ,N_{1}$*,* $l_{2,i}=1.5,\;i=1,2,\ldots ,N_{2}$ *and* $l_{3,i}=1.5,\;i=1,2,\ldots ,N_{3}$*, respectively, are the widths of their receptive fields. For the first neural network* ${\hat{\omega }}_{1}^{⊤}S_{1}({\overset{⌣}{Z}}_{1})$*, the centers of its receptive domain are* $\varpi_{1,1}=-0.7$*,* $\varpi_{1,2}=0$*,* $\varpi_{1,3}=0.7$*; for the second neural network* ${\hat{\omega }}_{2}^{⊤}S_{2}({\overset{⌣}{Z}}_{2})$*, the centers of its receptive domain are distributed in the grid* {−0.7,0,−0.7}×{−0.8,0.8}×{−0.3,0.3}×{−0.5,0.5}*; for the third neural network* ${\hat{\omega }}_{3}^{⊤}S_{3}({\overset{⌣}{Z}}_{3})$*, the centers of its receptive domain are distributed in the grid* {−0.7,0,−0.7}×{−0.7,0.7}×{−0.7,0.7}×{−0.5,0.5}×{−0.5,0.5}×{−0.5,0.5}.

**Remark** **9.***In order to avoid uncontrollable situations, our initial values are chosen to be within the state constraints, namely,* $x_{1}\in (-k_{b_{1}},k_{b_{1}})$*,* $x_{2}\in (-k_{b_{2}},k_{b_{2}})$*,* $x_{3}\in (-k_{b_{3}},k_{b_{3}})$*. Within the above range, the initial value can be chosen arbitrarily. Regarding other parameters, the case of setting is not unique. Since we only need to justify the theory presented in this paper, all it takes is the existence of a situation in which it holds. The specific setting of the parameters varies with the case of the dynamic equations.*

Figure 2 depicts the evolution of the system’s state variable, as time passes, each one converges and satisfies the preset state constraints. Figure 3 shows the curves of the virtual controller and the actual controller with time, all three show the process from fluctuation to convergence. The attack compensator’s changing curve is depicted in Figure 4. Figure 5 shows the system error variables’ change curve, which converges with time and follows the same general pattern as the system state variables and the controller. Figure 6 depicts the first-order filter’s variation curve for the controller’s output variables, and Figure 7 depicts the RBF neural network’s approximate variation curve. Using the control technique described above can successfully maintain the stability of the nonlinear system under unknown deception attacks and better satisfy the predefined full-state constraints, as can be shown when combined with the aforementioned simulation findings.

***Example*** ***2.***
*We introduce the chemical reactor recovery system in [44] to verify the practical applicability and theoretical correctness of the theoretical approach in this paper. As mentioned in [44], there is always some unnecessary waste in chemical reaction systems. In order to improve the utilization of resources, chemical reactor recycling systems were born. Their main operation is separation, separating raw materials from products which are later recycled through different paths. In the process of recycling, delays are inevitably generated. Moreover, during the operation of the whole device, it is also very vulnerable to external deception attacks that put the whole system in crisis.*


In this paper, the problem of adaptive stability control of a class of three-stage chemical reaction cycle systems with time delays is studied. The dynamics of a three-stage chemical reaction system are shown in [44].
(72)$$\left\{\begin{array}{c} {\dot{x}}_{1}=-({ℵ}_{A}+\frac{1}{{℘}_{A}})x_{1}-\frac{1}{{℘}_{A}}x_{1}(t-\tau_{1})+\frac{1-\partial_{A}}{{Κ}_{A}}x_{2}\\ {\dot{x}}_{2}=\frac{ϒ}{{Κ}_{B}}u-{ℵ}_{B}x_{2}-\frac{1}{{℘}_{B}}x_{2}^{2}-\frac{2℧}{{℘}_{B}}x_{2}+\frac{\partial_{B}}{{Κ}_{B}}x_{1}(t-\tau_{1})+\frac{\partial_{B}}{{Κ}_{B}}x_{2}(t-\tau_{2}) \end{array}\right.$$
where ${ℵ}_{A}={ℵ}_{B}=0.5$ are reaction constants, $\partial_{A}=\partial_{B}=0.5$ are the flow rates of recycle, ${Κ}_{A}={Κ}_{B}=0.5$ are the volume of reactors, ${℘}_{A}={℘}_{B}=2$ are the reactor residence times, ϒ=0.5 is the feed rate of this system, and the equilibrium point is $℧=\frac{7}{3}$. In addition to this, $x_{1}$ and $x_{2}$ are the reactor production stream components, which are similar to the state variables of the system.

If we convert (72) into the form of (1) and consider the deception attacks from outside, we will get the following one:(73)$$\left\{\begin{array}{c} {\dot{x}}_{1}=x_{2}+f_{1}(x_{1})+h_{1}(x_{1}(t-\tau_{1}))\\ {\dot{x}}_{2}=u+f_{2}({\bar{x}}_{2})+h_{2}({\bar{x}}_{2}(t-\tau_{2}))+\varphi_{a}(t,{\bar{x}}_{2})\\ {\overset{⌣}{x}}_{j}=x_{j}+\varphi_{j,s}(t,x_{j}) \end{array}\right.$$
where j=1,2 are the number of sensors, the unknown nonlinear terms are $f_{1}(x_{1})=-({ℵ}_{A}+\frac{1}{{℘}_{A}})x_{1}$ and $f_{2}({\bar{x}}_{2})=-{ℵ}_{B}x_{2}-\frac{1}{{℘}_{B}}x_{2}^{2}-\frac{2℧}{{℘}_{B}}x_{2}$, the unknown terms with time-delay are $h_{1}(x_{1}(t-\tau_{1}))=-\frac{1}{{℘}_{A}}x_{1}(t-\tau_{1})$ and $h_{2}({\bar{x}}_{2}(t-\tau_{2}))=\frac{\partial_{B}}{{Κ}_{B}}x_{1}(t-\tau_{1})+\frac{\partial_{B}}{{Κ}_{B}}x_{2}(t-\tau_{2})$. In addition to the system functions, we also have deception attacks of sensors as $\varphi_{1,s}=0.8+\cos(2t)$ and $\varphi_{2,s}=0.3+\sin(t)$, deception attack of actuator is $\varphi_{a}=\cos(t)\cos(x_{2}(t))x_{1}(t)$. The uncertainty of the system is reflected in the unknown nature of the $f_{i}({\bar{x}}_{i})$, $h_{i}({\bar{x}}_{i}(t-\tau_{i}))$ function.

According to the adaptive resilient control method designed above, we can obtain
(74)$$\alpha_{2}=-l_{1}{\overset{⌣}{e}}_{1}-k_{1}{\overset{⌣}{e}}_{1}-{\hat{\omega }}_{1}^{⊤}S_{1}({\overset{⌣}{Z}}_{1})$$
(75)$$u=-{\hat{\omega }}_{2}^{⊤}S_{2}({\overset{⌣}{Z}}_{2})+\frac{\alpha_{2}-z_{2}}{\nu_{2}}-(2k_{2}+l_{2}+{\left(2k_{2}+{\bar{k}}_{2}\right)}^{2}z_{2}^{2})\gamma_{2}-\frac{2k_{2}\gamma_{2}+{\left(2k_{2}+{\bar{k}}_{2}\right)}^{2}z_{2}^{2}\gamma_{2}}{1+2(k_{b_{2}}^{2}/\lambda_{2}^{2}-{\overset{⌣}{e}}_{2}^{2})}$$
(76)$${\dot{\hat{\omega }}}_{1}=\sigma_{1}({\overset{⌣}{e}}_{1}+\frac{{\overset{⌣}{e}}_{1}}{k_{b_{1}}^{2}/\lambda_{1}^{2}-{\overset{⌣}{e}}_{1}^{2}})S_{1}({\overset{⌣}{Z}}_{1})-\sigma_{1}a_{1}{\hat{\omega }}_{1}$$
(77)$${\dot{\hat{\omega }}}_{2}=\sigma_{2}(\gamma_{2}+\frac{{\overset{⌣}{e}}_{2}}{k_{b_{2}}^{2}/\lambda_{2}^{2}-{\overset{⌣}{e}}_{2}^{2}})S_{2}({\overset{⌣}{Z}}_{2})-\sigma_{2}a_{2}{\hat{\omega }}_{2}$$
(78)$${\dot{\hat{\delta }}}_{2}=(2k_{2}+l_{2}+{\left(2k_{2}+{\bar{k}}_{2}\right)}^{2}z_{2}^{2})z_{2}\gamma_{2}-(2k_{2}+{\bar{k}}_{2})z_{2}^{2}{\hat{\delta }}_{2}-\pi_{2}{\hat{\delta }}_{2}$$

The initial values of the system state variables are selected to be ${\bar{x}}_{2}={\left[0.4,-0.4\right]}^{⊤}$ for the system described in (73), it is very important to note that the initial values are chosen in such a way that they do not violate the state constraints, that is, $x_{1}\in (-k_{b_{1}},k_{b_{1}})$ and $x_{2}\in (-k_{b_{2}},k_{b_{2}})$. Their coefficients are $k_{1}=k_{2}=0.5$, $l_{1}=0.01$, $l_{2}=0.3$, ${\bar{k}}_{2}=0.2$, $\nu_{2}=0.003$, $\sigma_{1}=1.2$, $\sigma_{2}=1$, $a_{1}=0.008$, $a_{2}=0.01$, $\pi_{2}=1$, $k_{b_{1}}=k_{b_{2}}=0.75$. Neural networks ${\hat{\omega }}_{1}^{⊤}S_{1}({\overset{⌣}{Z}}_{1})$ and ${\hat{\omega }}_{2}^{⊤}S_{2}({\overset{⌣}{Z}}_{2})$ have $N_{1}=3$, $N_{2}=12$ neurons respectively, and $l_{1,i}=1.5,\;i=1,2,\ldots ,N_{1}$ and $l_{2,i}=1.5,\;i=1,2,\ldots ,N_{2}$, respectively, are the widths of their receptive fields. For the first neural network ${\hat{\omega }}_{1}^{⊤}S_{1}({\overset{⌣}{Z}}_{1})$, the centers of its receptive domain are $\varpi_{1,1}=-0.7$, $\varpi_{1,2}=0$, $\varpi_{1,3}=0.7$; for the second neural network ${\hat{\omega }}_{2}^{⊤}S_{2}({\overset{⌣}{Z}}_{2})$, the centers of its receptive domain are distributed in the grid {−0.7,0,−0.7}×{−0.8,0.8}×{−0.3,0.3}×{−0.5,0.5}.

Figure 8 shows the variation curves of the system state variables over time under the full-state constraints. Figure 9 shows the variation curves of the designed virtual controller and the actual controller over time. Figure 10 depicts the variation curves of the attack compensator over time. Figure 11 depicts the variation curves of the unknown nonlinearity of the RBF neural network approximation system. From the figures below, it can be seen that the curves all trend from violent vibration to gradual convergence, indicating that the method proposed in this paper can effectively combat unknown deception attacks under the condition of full-state constraints. It enables the chemical reactor recovery system to converge to the equilibrium state quickly even after the external spoofing attack.

## 6. Conclusions

For an unknown nonlinear system with unknown deception attacks and full-state constraints, an adaptive resilient control strategy based on neural networks is proposed. In the design process, the dynamic surface technique is used to improve the backstepping control strategy to reduce its computational complexity. The state variables of the system become unmeasurable due to the impact of the unknown deception attacks, so a novel compromised variables-based backstepping control method is used in this paper to overcome the problem of unavailable state variables. The unknown nonlinear terms are estimated using RBF neural networks, and an attack compensator is built to mitigate the impact of deception attacks. For the unknown time-delay terms, LKF is introduced to eliminate its influence on the controller design. The full-state constraint is an inevitable problem in practical engineering, so BLF is introduced to restrict the system state variables. Finally, the feasibility of the theory is confirmed by two simulation experiments. In future work, we can consider adding event-triggering techniques to achieve energy saving, or using finite-time adaptive control strategy to accelerate the convergence speed of the system, etc.

## Figures and Tables

**Figure 1 entropy-25-00900-f001:**
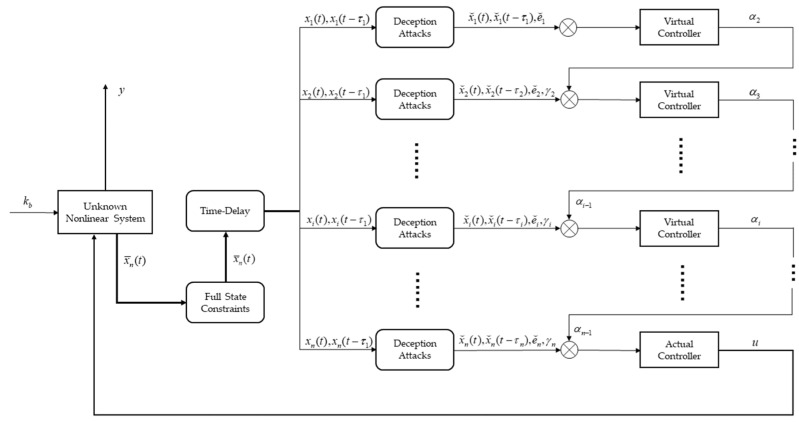
Block diagram of the method proposed in this paper.

**Figure 2 entropy-25-00900-f002:**
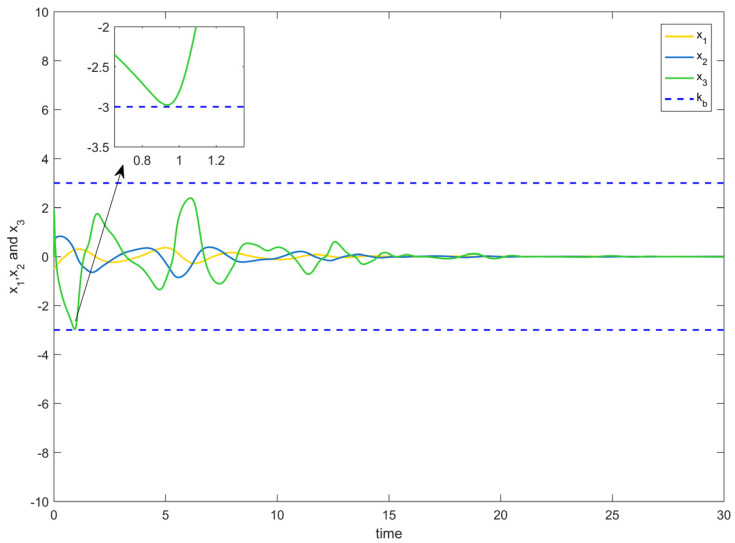
System state variable change trajectory of Example 1.

**Figure 3 entropy-25-00900-f003:**
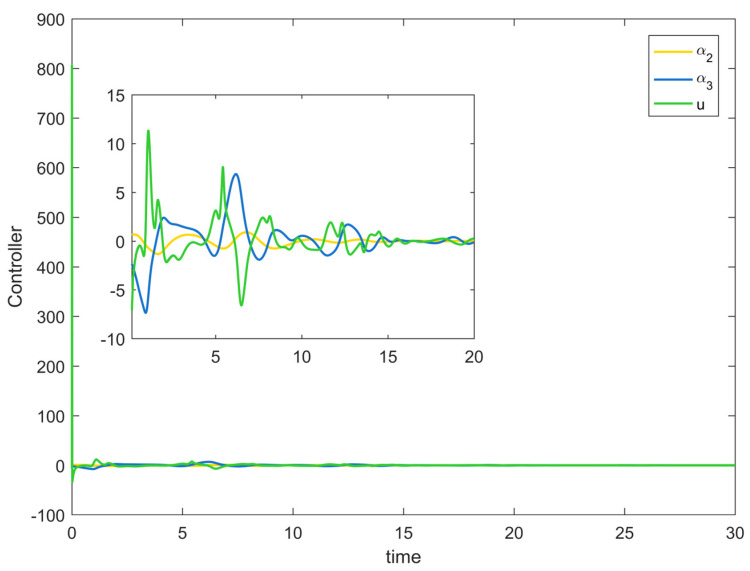
Controller change trajectory of Example 1.

**Figure 4 entropy-25-00900-f004:**
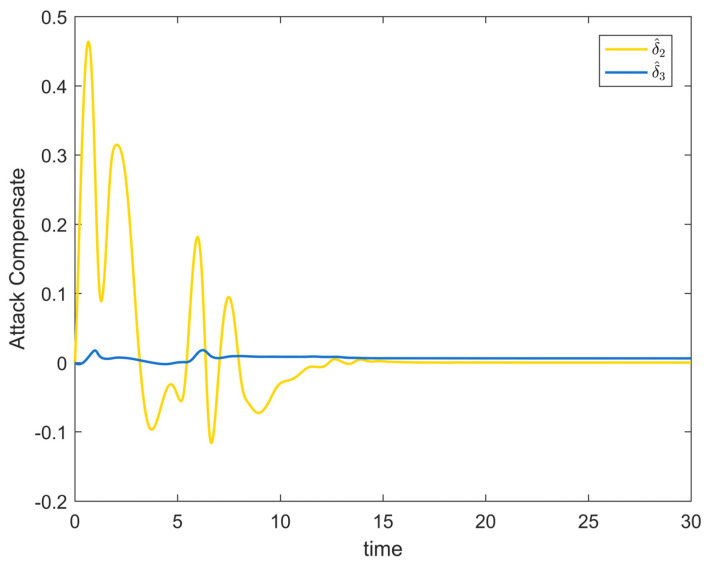
Attack compensator change trajectory of Example 1.

**Figure 5 entropy-25-00900-f005:**
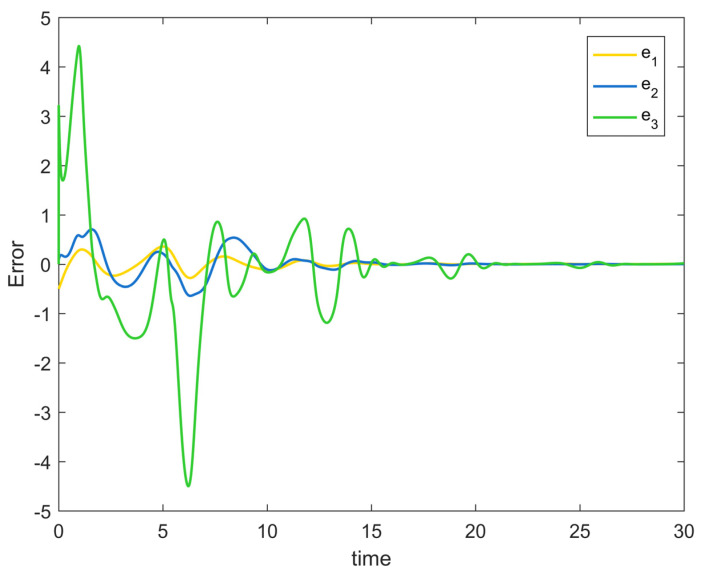
Error surface change trajectory of Example 1.

**Figure 6 entropy-25-00900-f006:**
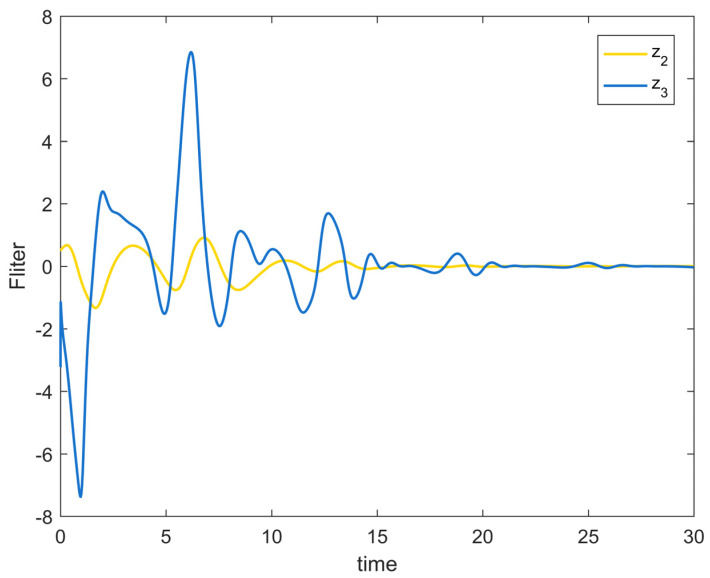
Output signal change trajectory of first-order filter of Example 1.

**Figure 7 entropy-25-00900-f007:**
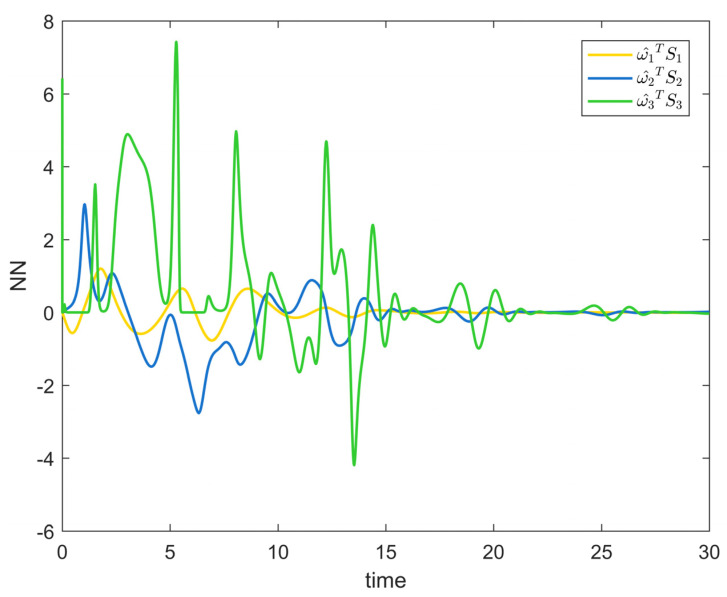
RBF neural network approximation curve of Example 1.

**Figure 8 entropy-25-00900-f008:**
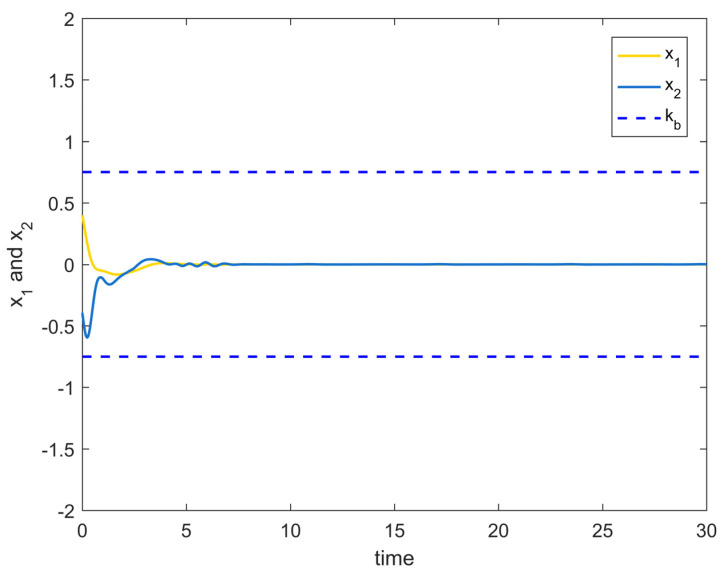
System state variables change trajectory of Example 2.

**Figure 9 entropy-25-00900-f009:**
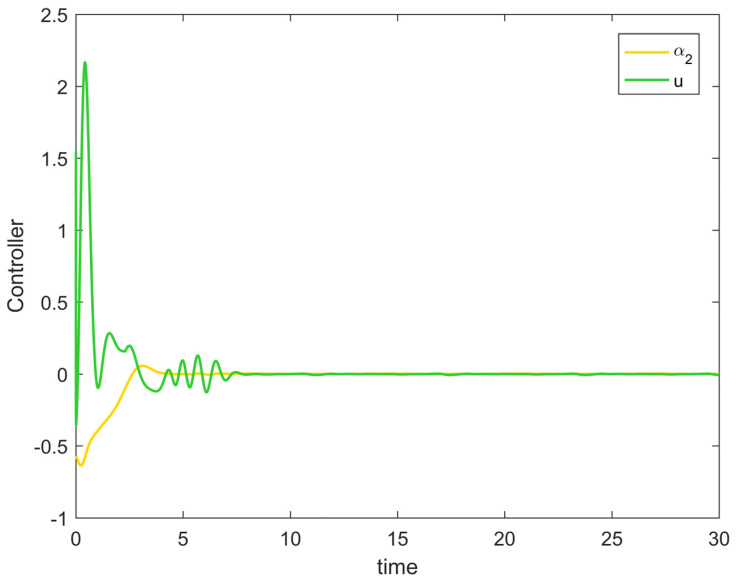
Controller change trajectory of Example 2.

**Figure 10 entropy-25-00900-f010:**
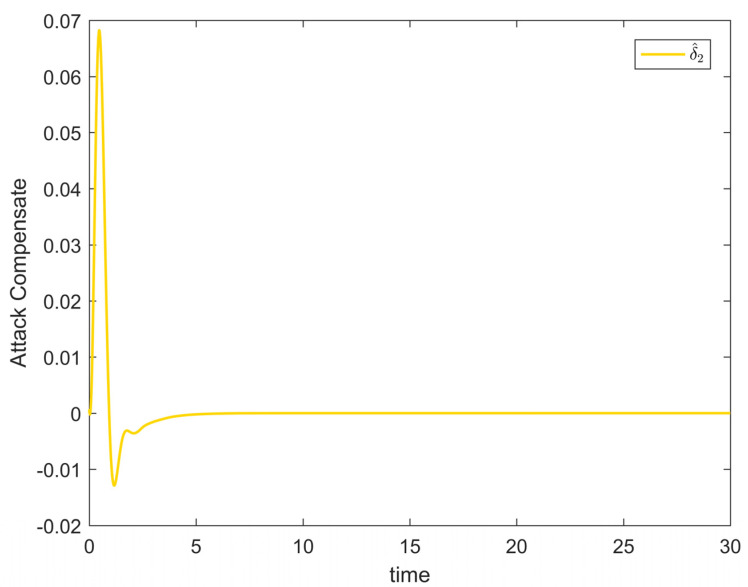
Attack compensator change trajectory of Example 2.

**Figure 11 entropy-25-00900-f011:**
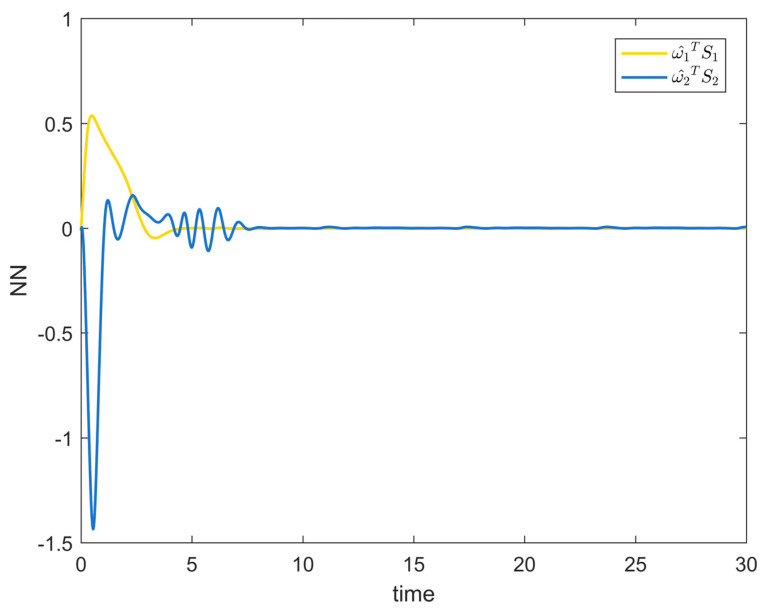
RBF neural network approximation curve of Example 2.

## Data Availability

No additional dataset is required for this article. Relevant simulation examples have been mentioned in Ref.

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
