# Peer review of "Adaptive Resilient Neural Control of Uncertain Time-Delay Nonlinear CPSs with Full-State Constraints under Deception Attacks"

_entropy, 2023, doi:10.3390/e25060900_

Round 1
Reviewer 1 Report
The paper develops a complex mathematical model of a non-linear cyber physical system with uncertain time delays, full state constraints, and is under unknown deception attacks. Towards this goal the paper introduces wide range of tools and techniques such as, backstepping control strategy, dynamic surface technique, radial basis function based neural networks, Lyapunov-Krasovskii function. The derived model is simulated and results are presented.
The paper starts out good by providing the reader with necessary description ensuring clarity throughout the introduction, problem formulation and methodology sections. In the fourth section which appears to be the central section of the entire paper, where the model is constructed leveraging different tools and techniques the clarity is lost due to the lack of context, structure and a bulk of equations (at least over 50 within this single section). Further, some symbols lack definition, for example, the psi in equation 15 is never defined. While the authors have dumped the equations in this section without context or structure to the section, the following section on simulation is extremely terse. This section lacks description of simulation experimental setup, description on the criteria used to select initial values and discussion on the selected values.
I would recommend a complete rewrite of Section 4 and 5. Section 4 must be structured and built block by block over several subsections, each providing enough context to the reader to make sense of the equations provided and to follow the derivations. Also, the entire derivation is not necessary to be provided in the section, such details can always be added as appendix. Section 5 should be more comprehensive detailing the transition from the model to simulation, the simulation method used, the scenario specification, justification to rationalize the initial values and communicate how the model captures/represents the salient modeled features.
Minor issue: on page 4 towards the end of the page, in the second bulleted point the (\lambda_j)(t) equation is wrong, needs to be fixed. Also, “Main Results” as the section title to Section 4 does not make much sense.
Author Response
Thank you very much for your comments, details of the response are in the pdf file.

Reviewer 2 Report
The manuscript presents an approach to provide stability of a non-linear cyber-physical system (CPS) under cyberattack. It is a good fit to the section “Complexity” of the journal.
The study builds on previous findings and presents results of potential interest to other researchers.
To increase the impact of the article, the authors need to provide a relevant real world example of a system by the general equation (1) or, better, the simulated example presented in equaltion (80). Otherwise, only very few readers would be interested to track all mathematical derivations included in the text. In addition, that would allow to understand the class of cases where this study is applicable. The authors need to elaborate on the limitations of the study. Providing stability is one of, and not necessarity the primary objective of control.
I understand that the terms “adaptive resilient control” have been used in the referenced literature, but while the approach to control and the controller can be designated as ‘adaptive’, resilient is (hopefully) the CPS.
In the introduction (p. 2) the article refers to three types of cyber attacks. The reference to the DOS attacks (and respective sources) is clearly irrelevant. The reference to replay attacks might be relevant,, but is not further pursued in the text. Hence, the authors need to focus on deception attacks and provide some real world examplaes (Stuxnet, etc.).
!! On the basis the “Author Contributions” (p. 20), I judge that only the first researcher needs to be listed as an author (who has provided the theoretical derivations, the simulations, and write the manuscript). None of the other three has made a substantive contribution to this manuscript.
The use of language is fairly good. Yet, it can be improved, e.g.:
· state-constrained constraints (p. 2)
· constrained by a variety of practical constraints (p. 3)
· “A novel adaptive resilient recursive control approach based on state variables damaged by deception attack signals” – what is damaged? ‘manipulated’?
Author Response

(The authors gave the same response as above.)

Reviewer 3 Report
The work is devoted to the adaptive control problem of a class of uncertain time-delay nonlinear cyber-physical systems (CPSs) with both unknown time-varying deception attacks and fullstate constraints.
Being scientifically and technically sound the work should address a series of comments in order to be accepted.
Comments:
1. Please describe clearly what is the main question addressed by the research. Is it the model or the method? I think that the work need developing the flowchart presenting the offered method.
2. Considering model (1) you need to evidence the existence of the solution of such type problem. What are the classes of the functions on the right side? Class of "nonlinear" functions is not enough to determine the properties of functions.
Do they need to be continuous, differential or bounded ones? Whar ae the restrictions for the functions of time delay in order that the solution exists?.
3. When analyzing other published material, you need to determine clearly what you work adds in the branch of security of CPS.
4. When differentiating the Lyapunov-Krasovski functions you need to replace "derivative" by "the upper right-hand derivative". 16, 19, 22 and so on throught the paper.
5. Subsection 4.3 on Stability should be improved. Namely, introduce the definitins of the stability studied? Global or local? Uniform? In the text present the main stability result. Discuss how to use it.
6. When introducing the model (1) please describe the uncertainties.
7. I recommend to apply the "attack" terminology used in the work DOI: 10.1109/ICCAS.2016.7832298 as "state" and "measurement" attacks (or analyze at the section on related works at least).
8. In the Section "Simmulation" describe the origin examples. Also technical particularities of modeling are of interest (software and libraries used, for NN).
The quality of English is good.
Author Response

(The authors gave the same response as above.)

Round 2
Reviewer 2 Report
Please use Arabic instead of Roman numbering the paragraph presenting the structure of the article (at the bottom of p. 4 in version2).
Author Response
Thank you for your suggestions for changes. Please see the attachment for detailed answers.

Reviewer 3 Report
All my comments have been addressed in the revised version of the manuscript. I recommend it for the publication
The English language is fine.
Author Response
Thank you for acknowledging the manuscript.
